# Invariant Causal Prediction with Local Models

**Alexander Mey**[1,2]         **Rui M. Castro**[1,3]

[1]Department of Mathematics and Computer Science, Eindhoven University of Technology, Eindhoven, The Netherlands
[2]ASML Research, Veldhoven, The Netherlands
[3]Eindhoven Artificial Intelligence Systems Institute (EAISI), Eindhoven University of Technology, Eindhoven, The Netherlands

## Abstract

We consider the task of identifying the causal parents of a target variable among a set of candidates from observational data. Our main assumption is that the candidate variables are observed in different environments which may, under certain assumptions, be regarded as interventions on the observed system. We assume a linear relationship between target and candidates, which can be different in each environment with the only restriction that the causal structure is invariant across environments. Within our proposed setting we provide sufficient conditions for identifiability of the causal parents and introduce a practical method called L-ICP (**L**ocalized **I**nvariant **Ca**usal **P**rediction), which is based on a hypothesis test for parent identification using a ratio of minimum and maximum statistics. We then show in a simplified setting that the statistical power of L-ICP converges exponentially fast in the sample size, and finally we analyze the behavior of L-ICP experimentally in more general settings.

## 1 INTRODUCTION

We consider the problem of identifying the causal parents of a target variable among a set of candidate variables, based only on observational data. As usual, causal inference and learning from observational data necessarily relies on assumptions. The main assumption used in this work is that data is collected in different *environmental* scenarios. An emblematic example is that of machine diagnostics, where we are monitoring several connected components of the machine. Different environments correspond, for example, to machines of the same model, but operating in different locations, different settings of a machine, or data collected in different points in time. If the system behaves differently across environments we talk about *heterogeneous* environments, and these can then be interpreted as accidental interventions on the system. The invariant causal prediction (ICP) principle, pioneered by Peters et al. [2016], is a particular way of using heterogeneous environments for causal discovery. It is based on the idea that the performance of any prediction model for the target variable should be invariant under interventions on the covariates, if and only if all covariates within the model are causal parents of the target. We extend upon that work by relaxing Peters et al. [2016] global linearity with a *local* linearity assumption, meaning that each environment is equipped with its own linear model. Relaxing the global model to local models carries a couple of interesting implications that we address in this paper. In particular, local models extend the scope of what might be considered an (accidental) intervention. While heterogeneity, as viewed in Peters et al. [2016], is always introduced by interventions on the covariate distributions, heterogeneous local models can be seen as informative interventions on the mechanisms within the system.

Consider, for example, the task of identifying key factors that influence fluctuations in stock prices and market volatility; this highlights the relevance of our extension. Within our framework, we first identify key times when important legislation concerning the stock market was enacted. We then choose our environments as the time intervals between these legislative actions. One may assume that a legislation intervenes on the distributions of the important factors, but also on the mechanism between those factors and stock prices/ market volatility. In our setting both types of (accidental) interventions are allowed and useful for causal discovery.

The paper is organized as follows: Section 2 contextualizes our contributions, positioning them relative to the related work. Section 3 formally introduces the setting, assumptions and our inference goals. In addition, it features further concrete examples showcasing the meaningfulness of the modeling assumptions. Section 4 introduces a meta-approach and characterizes sufficient conditions under which this approach can identify the causal parents of a target variable.

*Accepted for the 40th Conference on Uncertainty in Artificial Intelligence* (UAI 2024).

Driven by that knowledge Section 5 introduces L-ICP, a practical approach to detect causal parents based on observational data. This approach inherits many of the properties of the idealized meta-procedure, and we show in a simplified setting that the power of L-ICP converges exponentially fast, in the sample size, towards one. Section 6 numerically illustrates the performance of L-ICP, and we conclude the paper in Section 7 with a discussion and an outlook on future work.

## 2 RELATED WORK

The idea of relying on heterogeneous environments for causal discovery is not new, and was instrumental for the work in Peters et al. [2016]. This setup was also extended to sequential data in Pfister et al. [2019], while Heinze-Deml et al. [2017] investigate non-linear models by using conditional independence tests. Relaxations of the global linearity assumption towards local models have been also considered: Christiansen and Peters [2020] and Zhou et al. [2022] assume that the local structural parameters, i.e. the parameters of the assumed linear relationship between target and its parents, are related through a common (unobserved) variable. Huang et al. [2019, 2020] consider a temporal setting, where the local structural parameters change according to an auto-regressive model. In contrast to these works we consider a setting where the structural parameters can be radically different for the various environments.

The following works aggregate the information from different environments, without relying on heterogeneity assumptions, while rather non-Gaussian noise or non-linear relationship assumptions play a pivotal role: Osman et al. [2023] propose a method that finds causal structures and detects interventions using a minimum description length score on the causal factorization, which they define jointly over the different environments. They assume non-linear relationships, identifiability of the DAG up the Markov equivalence class and a low noise condition. Chen et al. [2021] assume a linear structural equation model (SEM), such that at least in some environments the corresponding DAG is identifiable from data. Shimizu [2012] proposes an extension of LiNGAM [Shimizu et al., 2006] to the multiple environment case. LiNGAM is a method that can find DAGs under a linearity and non-Gaussianity assumption. Finally, Mooij et al. [2020] developed the joint causal inference (JCI) framework, where environments (in the paper called contexts) are directly encoded as part of the structural causal model (SCM).

## 3 SETTING

In this work we consider a scenario with two observable quantities of interest: a target, denoted by $Y$, and a set of covariates $X := (X_1, \cdots, X_D)$ for $D \in \mathbb{N}$. Our overar-

ching goal is to identify which of the $D$ covariates are the causal parents of the target $Y$. We further assume to have access to $E \in \mathbb{N}$ different environments and in each environment we receive $n^e$ observations, so that $Y^e \in \mathbb{R}^{n^e}$ and $X^e \in \mathbb{R}^{n^e \times D}$ are respectively the target and covariate observations. With $X_{d,i}^e$ we indicate the entry of $X^e$ in the $d$-th row and $i$-th column, which corresponds to the $d$-th covariate of observation $i$. For $S \subseteq [D]$ we write $X_S^e \in \mathbb{R}^{n^e \times |S|}$ to indicate the sub-matrix of $X^e$ with columns given by $S$. We assume that for each $e \in [E]$ the structural equation of $Y^e$ is given by

$$Y^e := X^e \beta^e + \varepsilon^e, \tag{1}$$

where $\varepsilon^e \in \mathbb{R}^{n^e}$ is a zero-mean random perturbation (specified explicitly below) and $\beta^e \in \mathbb{R}^D$ is the column vector of structural parameters. In the following $\beta_d^e$ indicates the $d$-th entry of $\beta^e$. As we consider the relationship $Y^e := X^e \beta^e + \varepsilon^e$ to be a structural causal model (SCM) in the sense of Pearl [2016], we consider the set $S^{e,*} := \{d \in [D] \mid \beta_d^e \neq 0\} \subseteq [D]$ as the true causal parents of $Y^e$. Correspondingly we define the set of causal parents of $Y$ as $S^* := \bigcup_{e \in [E]} S^{e,*}$. Our inference goal of finding the causal parents of $Y$ is then equivalent to finding $S^*$. It is important to note the vectors $\beta^e$ can be radically different across environments. The following assumption formalizes our setting and further specifies the independence assumptions made:

**Assumption 1.** *Let $S^* \subseteq [D]$ be defined as above, such that $\beta_d^e = 0$ for all $d \notin S^*$ and $e \in [E]$. There exists a zero mean distribution $F^*$ such that for all $e \in [E]$ we have that*

$$Y^e = X^e \beta^e + \varepsilon^e \quad \text{with:}$$
$$\varepsilon_i^e \sim F^* \quad \text{for all } i \in [n^e] \tag{2}$$
$$\varepsilon_i^e \perp\!\!\!\perp \varepsilon_j^e \quad \text{for all } i \neq j$$
$$\varepsilon_i^e \perp\!\!\!\perp X_{S^*,i}^e \quad \text{for all } i \in [n^e]. \tag{3}$$

In the above $\perp\!\!\!\perp$ indicates statistical independence. We draw special attention to condition (2), i.e., the assumption that $\varepsilon_i^e \sim F^*$ for all $e \in [E]$ and $i \in [n^e]$. This is arguably the strongest assumptions in our setting, and might be relaxed as discussed in Section 7 by allowing the distribution to vary across environments. It ensures that the noise distribution is the same in all environments, which is a crucial property we test for within our methodology. There are many scenarios where it is nevertheless a very reasonable assumption, e.g., in monitoring settings, where it can embody sensing noise - see example below. We finally highlight that heterogeneity across environments $e \in [E]$ embodies both heterogeneity in the distributions of the covariates $X^e$ (as in Peters et al. [2016]) and the parameters $\beta^e$. While heterogeneity plays a central role, we show in Theorem 1 that under mild conditions parent identification is possible for almost all values of $(\beta^e)_{e \in [E]}$. To illustrate the setting we now describe

two scenarios, that further stress the meaningfulness of the heterogeneity assumptions.

**Finding causal parents of a disease.** Assume we want to find causes for a certain disease with data from different countries, playing the role of different environments. We collect data from plausible risk factors for the disease (e.g., diet, lifestyle, genetic variations, etc.). Very plausibly risk factors are heterogeneous across different countries, and thus provides a scenario in which our setting and the resulting methodology are applicable. Next to the heterogeneity in the risk factors, unobserved factors such as the quality of the health care may also introduce heterogeneity in the structural parameters: a negative health outcome, given all risk factors, is much more likely if the health care is poor. In that case it becomes necessary to have local models.

**Finding causes of a mechanism shift.** We are collecting dynamical data from a machine and observe that a target variable $Y$ starts to drift, and we want to find the cause for that. The underlying, unknown, cause is that a certain component of the machine is degrading over time. This degradation naturally provides heterogeneous environments, if we set our environments as different time intervals of the observations. Note that in this setting (2) is deemed quite reasonable and might embody sensor measurement noise.

## 3.1 ADDITIONAL NOTATION

Some further notation we use throughout the paper: A graph $G = ([D], \mathcal{E})$ is a tuple where indices in $[D]$ represent nodes and $\mathcal{E} \subseteq [D]^2$ represent directed edges between nodes, with the assumption that $(d, d) \notin \mathcal{E}$ for any $d \in [D]$. If $(d_1, d_2) \in \mathcal{E}$ we call $d_1$ a *parent* of $d_2$ and we call $d_2$ a *child* of $d_1$. We call $d_1$ an *ancestor* of $d_k$ and we call $d_k$ a *descendant* of $d_1$ if there exists a sequence $d_1, \cdots, d_k$ such that $(d_i, d_{i+1}) \in \mathcal{E}$ for $1 \leq i < k$. If $d_k$ is not a descendant of $d_1$ we call $d_k$ a *non-descendant* of $d_1$. A node without any child is called a *sink node*. Given a node $d \in [D]$ we respectively define $\mathbf{PA}(d)$, $\mathbf{AN}(d)$, $\mathbf{DE}(d)$, $\mathbf{NDE}(d)$ as the set of all parents, ancestors, descendants and non-descendants of $d$. Furthermore, we use $\mathbf{E}[\cdot]$, $\mathbf{V}[\cdot]$ and $\mathbf{C}[\cdot, \cdot]$ respectively as the expectation, variance and covariance operator. Finally we set $(\boldsymbol{X}, \boldsymbol{Y}) := \{(X^e, Y^e)\}_{e \in [E]}$.

# 4 ON THE IDENTIFIABILITY OF CAUSAL PARENTS

Ultimately, our goal is to identify, based on data, a set $\tilde{S} \subseteq [D]$ of variables deemed causal parents, that is ideally identical to $S^*$. Towards this goal we develop a test-based methodology ensuring $\tilde{S} \subseteq S^*$ with high probability. The methodology works by identifying sets of *plausible causal parents*, which are also often called invariant sets (of covariates) in the relevant literature. Roughly speaking, a subset

$S \subseteq [D]$ is *plausible* if it allows for a data generation model as described by Assumption 1, when $S$ takes the place of $S^*$. The inferred set of causal parents $\tilde{S}$ is then the intersection of all plausible sets. In the following section we formalize those concepts and show that we can control the false positive discoveries, so that $\tilde{S} \subseteq S^*$ with high probability.

## 4.1 CONTROL OF FALSE POSITIVES

Given a set $S \subseteq [D]$ consider the following null hypothesis:

$$\tilde{H}_{0,S} : \begin{cases} \exists \text{ a distribution } F \text{ and } \gamma^e \in \mathbb{R}^{|S|}, \text{ s.t.} \forall e \in [E]: \\ Y^e = X_S^e \gamma^e + r^e \text{ and } \forall i, j \in [n^e], i \neq j: \\ r_i^e \sim F, r_i^e \perp\!\!\!\perp r_j^e, r_i^e \perp\!\!\!\perp X_{S,i}^e . \end{cases}$$

We note that $\tilde{H}_{0,S}$ corresponds to Assumption 1 when $S^*$ is replaced by $S$, which in particular implies that under Assumption 1 $\tilde{H}_{0,S^*}$ is true. However, it is not clear how to build a practical test based on $\tilde{H}_{0,S}$ that is also powerful against meaningful alternatives. For that reason we move towards a weaker but more practical formulation. Let

$$\tilde{\beta}_S^e = \mathbf{E}[(X_S^e)^t X_S^e]^\dagger \mathbf{E}[(X_S^e)^t Y^e],$$

where $A^\dagger$ denotes the generalized Moore-Penrose inverse of a matrix $A$ [Penrose, 1955], with the convention that $\tilde{\beta}_\emptyset^e = 0$. Formulate the following relaxation of $\tilde{H}_{0,S}$:

$$H_{0,S} : \begin{cases} \exists \text{ a distribution } F \text{ such that for all } e \in [E]: \\ Y^e = X_S^e \tilde{\beta}_S^e + r^e \text{ and } \forall i \in [n^e] : r_i^e \sim F . \end{cases}$$

The following lemma, proven together with all other formal statements in Section A of the supplementary material, establishes the relation between $H_{0,S}$ and $\tilde{H}_{0,S}$:

**Lemma 1.** *If $\tilde{H}_{0,S}$ is true then so is $H_{0,S}$.*

Suppose we have access to a collection of tests corresponding to the above null hypothesis $H_{0,S}$. Specifically, given the observations $(\boldsymbol{X}, \boldsymbol{Y})$ let $\phi_S(\boldsymbol{X}, \boldsymbol{Y}) \in \{0, 1\}$ be a test function, such that $\phi_S(\boldsymbol{X}, \boldsymbol{Y}) = 1$ indicates we reject $H_{0,S}$. We know that $\tilde{H}_{0,S^*}$ holds by Assumption 1, and thus Lemma 1 implies $H_{0,S^*}$ also holds, and we expect that with high probability $\phi_{S^*}(\boldsymbol{X}, \boldsymbol{Y}) = 0$. With this in mind we view all $S$ for which $\phi_S(\boldsymbol{X}, \boldsymbol{Y}) = 0$ as a *plausible* set, and naturally define the estimator $\tilde{S}$ of $S^*$ as

$$\tilde{S} := \bigcap_{S : \phi_S(\boldsymbol{X}, \boldsymbol{Y})=0} S. \tag{4}$$

This definition of the parent estimator $\tilde{S}$ ensures control over false discoveries:

**Proposition 1.** *Let $\alpha \in (0, 1)$. Consider a class of test functions $\phi_S$ for all $S \subseteq [D]$ that satisfies $P[\phi_{S^*}(\boldsymbol{X}, \boldsymbol{Y}) = 1 \mid H_{0,S^*} \text{ holds}] \leq \alpha$. Then we have $\tilde{S} \subseteq S^*$ with probability of at least $1 - \alpha$.*

This result, which is essentially Theorem 1 from Peters et al. [2016], is a simple consequence of the fact that $P[\phi_{S^*}(\boldsymbol{X}, \boldsymbol{Y}) = 1 \mid H_{0,S^*}$ holds$] \leq \alpha$. Note that for this result it suffices to guarantee good behavior of the testing procedure for the *true* set $S^*$.

## 4.2 CONTROL OF FALSE NEGATIVES

Proposition 1 guarantees $\tilde{S} \subseteq S^*$ with arbitrarily high probability. In other words, we are guaranteed to not include non-causal parents with high probability. However, that can be trivially obtained for the choice $\tilde{S} = \emptyset$. Naturally, we ask under which assumptions one can have a class of tests that ensure that $\tilde{S} = S^*$ with high probability as well. The answer to this question is significantly more intricate, and depends crucially on how much information is present in the data. To shed some light on this matter we consider a *population* setting, effectively focusing on scenarios where one has an arbitrarily large amount of data in each environment.

Specifically, suppose one has access to $\tilde{\beta}_S^e$ for any $e \in [E]$ and $S \subseteq [D]$. Theorem 1 below shows that, within a fairly general class of structural equation models for the covariates $X^e$, the proposed approach can identify $S^*$, and only in rather pathological combinations of parameters issues might arise. The class of structural equation models for which we can show identifiability is given through the following assumption:

**Assumption 2.** *To simplify notation, we introduce the index $y := D + 1$ and define $X_y^e := Y^e$. We assume that $E \geq 2$ and for all $e \in E$ it holds that $n^e > 0$. Furthermore for at least one $e \in E$ there exists an acyclic graph $G = ([D] \cup y, \mathcal{E})$ such that for $d \in [D]$ the structural equation of $X_d^e$ has the form*

$$X_d^e := f_d^e(X_{\boldsymbol{PA}(d)}^e) + \delta_d^e,$$

*where $f_d^e$ is a polynomial of finite degree in $X_y^e = Y^e$ if $y \in \boldsymbol{PA}(d)$, but otherwise arbitrary. More specifically in that case $f_d^e$ has the form*

$$f_d^e(X_{\boldsymbol{PA}(d)}^e) = \sum_{k=0}^{K} (Y^e)^k g_k^e \left( X_{\boldsymbol{PA}(d) \setminus y}^e \right),$$

*where $K < \infty$ and the functions $g_k^e$ are arbitrary. In the above $\delta_d^e = (\delta_{d,1}^e, \cdots, \delta_{d,n^e}^e) \sim \mathcal{D}_d^e$ is a random noise vector such that*

$$\forall : i \in [n^e], u \in \boldsymbol{NDE}(d) : \quad \delta_{d,i}^e \perp\!\!\!\perp X_{u,i}^e \qquad (5)$$

*where $\mathcal{D}_d^e$ is a distribution such that $\Delta_{d,i}^e := \boldsymbol{V}(\delta_{d,i}^e) > 0$ for all $i \in [n^e]$. We define $\Delta^e \in (0, \infty)^{D \times n^e}$ to be the matrix with the $(d,i)$-th entry given by the variance $\Delta_{d,i}^e$. Finally, we assume that for all $e \in [E], d \in [D]$ and $i \in [n^e]$ the covariate $X_{d,i}^e$ has finite variance.*

Informally speaking the noise terms $\delta_d^e$ ensure that each covariate introduces unique information and prevent that the causal parents $X_{S^*}^e$ lie in the column space of any other subset of variables $X_S^e$ with $S^* \not\subseteq S$. The explicit variances $\Delta_{d,i}^e$ are needed due to our proof technique, but we highlight that we do not require $\Delta_{d,i}^e$ to be heterogeneous in the environments. The polynomial dependence on $Y^e$ simplifies our proof, could, however, be replaced by other regularity assumptions on $f_d^e$. With this in hand we are ready to state our main identifiability result:

**Theorem 1.** *Let $S \subset [D]$ such that $S^* \not\subseteq S$ and take any two environments $e, v \in [E]$ with $e \neq v$, such that environment $e$ fulfills the data generation mechanism from Assumption 2 with variances $\Delta^e$. Suppose Assumption 1 holds with parameters $\beta^e, \beta^v$. Then there exists a set $M_0 \subset \mathbb{R}^{|S^*| \times |S^*|} \times (0, \infty)^{D \times n^e}$ with Lebesgue measure zero, such that if*

$$(\beta^v, \beta^e, \Delta^e) \notin M_0$$

*it is guaranteed that $H_{0,S}$ is false.*

***Sketch Proof.*** *We use the polynomial relationships of Assumption 2 to show that the variances of the residuals, when regressing $Y^e$ onto the variables from $S$, are a ratio of polynomials with respect to a distinguished structural parameter $\beta_u^e$ for $u \in [D]$. The hypothesis $H_{0,S}$ can only be true if the variances of those residuals are equal in all environments. As the variances are a ratio of finite polyonimals with respect to $\beta_u^e$, this equality can only be established for finitely many choices of $\beta_u^e$, leading to the null set $M_0$.*

Informally the above theorem states that the parent set $S^*$ can always be identified, with the exception of very specific (pathological) parameter combinations $(\beta^v, \beta^e)$. An interpretation of this statement is that within our framework it is possible to identify causal relationships for the vast majority of (accidental) interventions on the mechanisms. An interesting consequence is that one can in principle recover a complete causal graph, and not only the causal parents of a chosen target. For that we mainly require that the noise distributions of all covariates are homogeneous, and heterogeneity is only introduced through changing structural parameters. With that, every covariate can take the role of the target, and all of our assumptions are still fulfilled. In Section 6.1 we illustrate this by using the proposed methodology to recover a causal graph based on data from a nonlinear dynamical system. Viewing small time-intervals as the environments, the local model can be viewed as a local approximation to the system, and the heterogeneity of this approximation is introduced by the non-linearity of the system.

# 5 PROPOSED APPROACH AND FINITE SAMPLE RESULTS

Theorem 1 relies on the values $\tilde{\beta}_S^e$, which are not observable, and we thus replace $\tilde{\beta}_S^e$ by suitable estimates based on the observed data. A natural choice is the solution of a generalized least-squares problem

$$\hat{\beta}_S^e := ((X_S^e)^T X_S^e)^\dagger (X_S^e)^T Y^e,$$

where we set $\hat{\beta}_\emptyset^e = 0$. Define also the residuals

$$r_S^e := Y^e - X_S^e \hat{\beta}_S^e.$$

To make use of our meta-procedure (4) we still need to define the set of tests $\phi_S$, and to facilitate this we make the following additional assumption.

**Assumption 3.** *The random noise variables $\varepsilon_i^e$ are sampled from a Gaussian distribution with zero mean and unknown variance $\sigma_Y^2$. Furthermore, we assume the following independencies for all $e, v \in [E]$ with $e \neq v$*

$$\varepsilon_i^e \perp\!\!\!\perp \varepsilon_j^v \quad for \ all \quad i \in [n^e], j \in [n^v]$$
$$\varepsilon_i^e \perp\!\!\!\perp \varepsilon_j^e \quad for \ all \quad i, j \in [n^e], i \neq j.$$

While this is a strong distributional assumption on the observation noise, it serves primarily as a driver to propose a concrete testing methodology. Section 6 examines the robustness of the methodology to violation of this assumption, while in Section 7 we discuss possible ways to extend the methodology towards non-Gaussian noise.

With all the ingredients in hand, we define the following test statistic:

$$T_S(\boldsymbol{X}, \boldsymbol{Y})$$
$$:= \begin{cases} \dfrac{\min_{e \in [E]} \|r_S^e\|_2^2}{\max_{e \in [E]} \|r_S^e\|_2^2} & \text{if} \quad \exists e \in [E]: \ \|r_S^e\|_2 > 0 \\ \infty & \text{otherwise} \end{cases}.$$

Under Assumption 3 we know that $\sigma_Y^2 \|r_{S^*}^e\|_2^2$ are all chi-squared distributed (note we are considering $S^*$), and the number of degrees of freedom depends only on the properties of the Gram matrix. Importantly, the scaling $\sigma_Y^2$ is the same for all $e \in [E]$, which implies that the distribution of $T_{S^*}(\boldsymbol{X}, \boldsymbol{Y})$ is not a function of $\sigma_Y^2$. Therefore, we can easily calibrate a test based on $T_S(\boldsymbol{X}, \boldsymbol{Y})$ using only observable quantities. This test statistic is motivated by the problems of sparse testing [Ingster, 1997, Donoho and Jin, 2004, Stoepker et al., 2022], and it targets scenarios where we expect evidence for rejection of the null hypothesis to be present in few environments.

To calibrate a test based on this statistic we first define $Z_S^e$ for $e \in [E]$ as jointly independent chi-squared random

---

**Input:** $(\boldsymbol{X}, \boldsymbol{Y}), \alpha$. In order: observations, confidence level
**Output:** $\tilde{S}$, the estimated causal parents

> $\tilde{S} = \emptyset$
> For all $S \subseteq [D]$ :
> > If $\phi_S(\boldsymbol{X}, \boldsymbol{Y}, \alpha) = 0$:
> > > If $\tilde{S} = \emptyset$:
> > > > $\tilde{S} = S$
> > > Else:
> > > > $\tilde{S} = \tilde{S} \bigcap S$
> Return $\tilde{S}$

Algorithm 1: Our proposed method L-ICP.

variables, respectively with $n^e - \text{rank}((X_S^e)^T X_S^e)$ degrees of freedom (zero degrees of freedom correspond to $Z_S^e = 0$). Given $(\boldsymbol{X}, \boldsymbol{Y})$ these variables are also independent of all the other quantities and we define the test $\phi_S$ as

$$\phi_S(\boldsymbol{X}, \boldsymbol{Y}, \alpha) := \tag{6}$$
$$\begin{cases} 1 & \text{if } P\left( T_S(\boldsymbol{X}, \boldsymbol{Y}) > \dfrac{\min_{e \in [E]} Z_S^e}{\max_{e \in [E]} Z_S^e} \ \middle| \ \boldsymbol{X}, \boldsymbol{Y} \right) \leq \alpha \\ 0 & \text{otherwise} \end{cases}.$$

While the distribution of $\min_{e \in [E]} Z_S^e / \max_{e \in [E]} Z_S^e$ is not easy to characterize analytically, we can easily generate samples from it, so calibration by Monte-Carlo simulation is extremely simple and convenient. The overall procedure, called L-ICP, is described in Algorithm 1. Note that this description may seem computationally prohibitive, due to the complexity of the for-loop. In Section 7 issue further remarks on this.

The correct coverage of this procedure is a direct consequence of the guarantees already provided for the meta-procedure in Section 4.2.

**Proposition 2.** *Consider Assumptions 1 and 3 , and let $\alpha \in (0, 1)$. Then*

$$\mathbf{P}(\tilde{S} \not\subseteq S^*) \leq \alpha,$$

*for $\tilde{S}$ being the output of Algorithm 1.*

The probability of including a false positive parent is relatively easy to understand as this does not depend on anything other than Assumptions 1 and 3. Controlling false negative discoveries, so controlling the probability that $\phi_S(\boldsymbol{X}, \boldsymbol{Y}) = 0$ for $S \subseteq [D]$ with $S^* \not\subseteq S$, becomes much more complex. The results in the following section try to shed some light into this within a simplified setting.

## 5.1 FINITE SAMPLE RESULTS

To provide finite sample results on the power of L-ICP we make strong assumptions on the data generation procedure.

**Assumption 4.** *Let the number of environments $E$ be even, and let $[E_1], [E_2]$ denote two index sets for two types of environments such that $[E] = [E_1] \dot\cup [E_2]$ with $|[E_1]| = |[E_2]| = \frac{E}{2}$. In each individual environment we observe $n > D$ observations (i.e., $\forall e \in [E]$ $n^e = n$). For all $v \in [E_1]$ and $d \in [D]$ the $d - th$ covariate $X_d^v \in \mathbb{R}^n$ is an $n$-sample from $\mathcal{N}(\mu_d^v, \sigma_d^v)$, a normal distribution with mean $\mu_d^v$ and standard deviation $\sigma_d^v$. The samples are independent of each other and independent of the other covariates of the environment $v$. Similarly, for all $w \in [E_2]$ we sample $X_d^w$ from $\mathcal{N}(\mu_d^w, \sigma_d^w)$ with the same independence assumptions. We assume that there exists $\beta^1, \beta^2 \in \mathbb{R}^D$ such that $\beta^v = \beta^1$ and $\beta^w = \beta^2$ for all $v \in [E_1], w \in [E_2]$.*

**Remark 1.** *While in the setting above inverse matrices $(X_S^{wT} X_S^w)^{-1}$ and $(X_S^{vT} X_S^v)^{-1}$ exist for $v \in [E_1]$ and $w \in [E_2]$ and any $S \subseteq [D]$ with probability one, we consider for simplicity only the case that they exist.*

While this independence assumption is certainly strong and unrealistic, this setting is already non-trivial: without further assumptions, one cannot distinguish cause and effect, see for instance Example 1 in Mooij et al. [2016]. While in effect we assume that there are only two types of environments to simplify the analysis, we note that our algorithm does not have access to this information.

In our results we want to characterize the probability to miss a causal parent, which happens if $\phi_S(\boldsymbol{X}, \boldsymbol{Y}) = 0$ for any $S \subseteq [D]$ with $S^* \not\subseteq S$. To understand the distribution of $\phi_S(\boldsymbol{X}, \boldsymbol{Y})$ we need a notion of how much environment heterogeneity is introduced by the covariates in $U := S^* - S$, as this is the driver for L-ICP to identify causal parents. Let $(\sigma_Y)^2$ be the variance of the target noise $\varepsilon^e$ and define

$$\tilde{I}_S := \frac{\sum_{u \in U} (\beta_u^1)^2 (\sigma_u^v)^2 + (\sigma_Y)^2}{\sum_{u \in U} (\beta_u^2)^2 (\sigma_u^w)^2 + (\sigma_Y)^2}. \tag{7}$$

Then the residual heterogeneity in the environments, when we model the target $Y$ with covariates from $S$, is carried in the quantity $I_S$ defined as

$$I_S := \min\left\{ \tilde{I}_S, \frac{1}{\tilde{I}_S} \right\}. \tag{8}$$

Note that $0 < I_S \leq 1$ and small values of $I_S$ indicate a higher environment heterogeneity. Note in particular that $I_S = 1$ if for all $u \in U$ we have $(\beta_u^1)^2 = (\beta_u^2)^2$ and $(\sigma_u^v)^2 = (\sigma_u^w)^2$. The following result presents bounds on the false negative probability in terms of the sample size $n$ and the heterogeneity parameter $I_S$. We already disclaim that the result treats the effect of the number of environments $E$

crudely, and due to our proof technique it is actually vacuous for the case that $E \to \infty$. We instead chose to analyze the setting $E \to \infty$ in isolation, and a corresponding result is presented afterwards.

**Theorem 2.** *For $S \subset [D]$, with $S^* \not\subseteq S$ define $I_S$ as above and set $k := n - |S|$. If Assumptions 1,3, 4 are true and $I_S < 1$, then for any confidence level $\alpha \geq 0$ it holds that*

$$\mathbf{P}(\phi_S(\boldsymbol{X}, \boldsymbol{Y}) = 0) \leq$$

$$\frac{4E}{\alpha} \left( \left( \frac{1}{(I_S)^{\frac{1}{4}}} e^{(1 - 1/(I_S)^{\frac{1}{4}})} \right)^{\frac{k}{2}} + \left( (I_S)^{\frac{1}{4}} e^{(1 - (I_S)^{\frac{1}{4}})} \right)^{\frac{k}{2}} \right).$$

*This means that the probability to accept $S$ falsely as a plausible set drops exponentially fast in $k$, since $(ce^{1-c}) < 1$ for any $c \neq 1$.*

The proof of this and the following result is presented in Section A. We still owe the reader a result elucidating the case $E \to \infty$:

**Theorem 3.** *Let $S \subseteq [D]$, with $S^* \not\subseteq S$, $I_S$ as defined above, and $k := n - |S|$. To emphasize the dependence of the data on $E$ we write now $(\boldsymbol{X}_E, \boldsymbol{Y}_E) = \{(X^e, Y^e)\}_{e \in E}$. If Assumptions 1,3,4 hold then for any $\alpha \geq 0$ we have that*

$$\lim_{E \to \infty} \mathbf{P}(\phi_S(\boldsymbol{X}_E, \boldsymbol{Y}_E) = 0) \leq \frac{1}{\alpha} \frac{2(I_S)^{\frac{k}{2}}}{2(I_S)^{\frac{k}{2}} + 1}.$$

*If we further assume the collection of random variables $\{X_{d_1, i_1}^{e_1}, X_{d_2, i_2}^{e_2}\}$ for $e_1 \in [E_1], e_2 \in [E_2]$ and $i_1, i_2 \in [n]$ to be mutually independent then for any $\alpha < \frac{(I_S)^{\frac{k}{2}}}{(I_S)^{\frac{k}{2}} + 1}$ it holds that*

$$\frac{1}{1 - \alpha} \left( \frac{(I_S)^{\frac{k}{2}}}{(I_S)^{\frac{k}{2}} + 1} - \alpha \right) \leq \lim_{E \to \infty} \mathbf{P}(\phi_S(\boldsymbol{X}_E, \boldsymbol{Y}_E) = 0).$$

Comparing Theorem 2 and 3 we make the observation that the dependence of the bound on $I_S$ and $k$ is very similar, the biggest difference being that Theorem 2 loses a factor of $\frac{1}{4}$ in the exponent of $I_S$ compared to Theorem 3. This may, however, very well be due to the proof technique of Theorem 2, in particular the use of Lemma 2. More importantly, Theorem 3 shows that the false acceptance probability does not necessarily converge to 0 for increasing $E$. This indicates a potentially complicated relationship between the number of available environments $E$ and the performance of L-ICP. To study this relation further, we conduct experiments in the following section.

## 6 EXPERIMENTAL RESULTS

The code generating all results from this section is accessible through https://github.com/AlexanderMey/causal-local-linear/tree/main/UAI-code.

We now perform a range of experiments to further shed light on the performance of L-ICP under model-misspecification. Furthermore, we contrast L-ICP with joint LiNGAM [Shimizu, 2012], ICP [Peters et al., 2016] and PCMCI [Runge et al., 2019]. For the implementation of the tests $\phi_S$ in the following experiments we generate, unless stated otherwise, $B = 1000$ samples from $\min_{e \in [E]} Z_S^e / \max_{e \in [E]} Z_S^e$ to compute the $p$-values for the tests in L-ICP. We generate data from a linear structural equation model with varying noise distributions and $|S^*| = 2, D = 6$, see Section B in the supplements for further details. In all experiments we generate data over 300 independent runs, collect the estimated causal parents in each run, and report how often the method missed a causal parent (false negative rate) and how often the method returned a non-parent (false positive rate). More precisely, let $\tilde{S}_r$ be the estimate of $S^*$ in run $r$. The false negative rate is given by $\frac{1}{300} \sum_{r=1}^{300} \mathbf{1}\left\{S^* \setminus \tilde{S}_r \neq \emptyset\right\}$ and the false positive rate by $\frac{1}{300} \sum_{r=1}^{300} \mathbf{1}\left\{\tilde{S}_r \setminus S^* \neq \emptyset\right\}$. We also report error bars, which are computed as a 95 percent Clopper-Pearson confidence interval [Clopper and Pearson, 1934]. Unless stated otherwise, all tests of L-ICP are done with target level $\alpha = 0.1$.

**Effects of Non-Normal Noise.** To calibrate L-ICP we make the normal noise Assumption 3 and we first investigate the impact on the performance of L-ICP when this is violated. As our test is based on minimal and maximal statistics, we expect that a misspecification of the tail distribution has the biggest impact on the performance. We thus generated data with three different noise models: normal noise, for a baseline comparison, uniform noise and Student-t distributed noise, where the standard deviation of the noise is kept at $1.1$.[1] This results in approximately 11.5 degrees of freedom for the Student-t distribution. In Figure 1 we show that under uniform noise, L-ICP is more conservative and under the Student-t noise it is less conservative, where the false positive rate exceeds at times the target threshold of $0.1$. The results for the false negative rates are similar and shown in Figure 2. We highlight that the performance gap of the false positive rate and the false negative rate under Student-t noise has the same reason: if the correct set $S^*$ is not accepted as a plausible set, we cannot guarantee that $\tilde{S} \subseteq S^*$, but if $S^*$ is not a plausible set, it also becomes more likely that $S^* \not\subseteq \tilde{S}$. While this can be addressed by adjusting $\alpha$, as also confirmed with an additional experiment in Section B.1, it is not clear what the correct adjustment is, and for that we need to investigate ways to calibrate the method without the normality assumption as further discussed in Section 7.

[1]Note that the standard deviation for standard Student-t distribution is always larger than 1, approaching that value in the limit of the number of degrees of freedom.

---

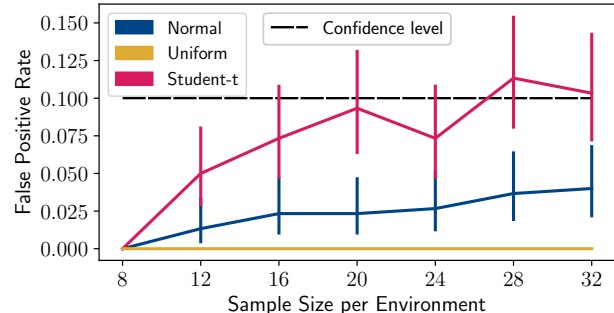

Figure 1: The behavior of the false positive rate of L-ICP under a misspecified noise model.

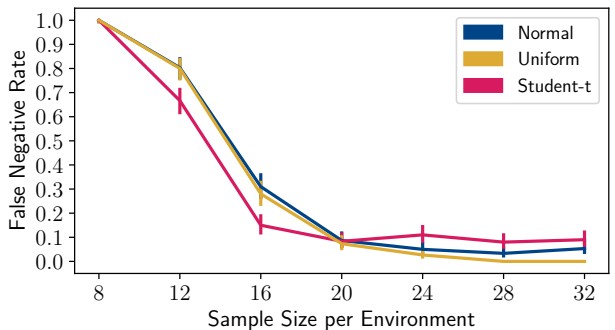

Figure 2: The behavior of the false negative rate of L-ICP under a misspecified noise model.

**Comparison with ICP.** As our method is an extension of ICP, we now highlight the main differences and showcase some consequences of those in three simple experiments. The main difference of ICP is that it additionally assumes that $\beta^v = \beta^w$ for all $v, w \in [E]$. This restriction of course allows for a better parameter estimation, as we may pool data, and also for different hypothesis tests. In particular they additionally test (Method II in their paper) if the mean of the residuals is identical in all environments. In our current formulation this is not meaningful, as our residuals have a vanishing mean in each environment. Furthermore, while we test for differences in the minimum and maximum, ICP loops over all environments $e$ and tests if the mean and variance of $e$ is the same as the means and variances in the other environments. They then correct for this multiple test with a Bonferroni correction.

Considering the conceptual and practical differences in ICP and L-ICP we propose three experiments to test the practical implications. The first two experiments satisfy ICPs additional restriction that $\beta^v = \beta^w$ for all $v, w \in [E]$. In the first experiment 99 of a total 100 environments follow the same data generation procedure, so the heterogeneity is *sparse* in the environments. In the second setting the variance of the covariates is randomly sampled for 100 environment, so the heterogeneity is *dense* in the environments. In the last

| | False Positive Rate | | False Negative Rate | |
|---|---|---|---|---|
| **Data/Method** | ICP | L-ICP | ICP | L-ICP |
| **Dense** | **0.003** | 0.077 | 0.337 | **0.243** |
| **Sparse** | **0.007** | 0.107 | **0.503** | 0.703 |
| **ICP Violated** | **0.007** | 0.07 | 0.967 | **0.033** |

Table 1: A comparison of L-ICP and ICP in settings where the heterogeneity is dense/sparse in the environments and when the structural parameters $\beta^e$ are changing across environments, and thus violating one of ICPs assumptions.

setting the additional restriction of ICP is *violated*. While the complete description of the data generation can be found in Section B, the results are shown in Table 1. We notice that ICP is more conservative, leading to a false positive rate which is quite below the set threshold of $0.1$. This naturally results in a loss of power of ICP. In the dense case L-ICP achieves a lower false negative rate, even though the assumptions of ICP are entirely met. Curiously, in the sparse setting ICP achieves a lower false negative rate, despite the fact that our test targets sparse heterogeneity. This seems to be related to the specific test ICP relies on, together with pooling data from all environments to estimate the parameters $\beta$. Finally, as expected, ICP fails to produce meaningful results if the assumption that $\beta^v = \beta^w$ for all $v, w \in [E]$ is violated.

**Comparison with LiNGAM.** We now highlight the strength and pitfalls of L-ICP, while we compare its performance to a version of LiNGAM that can receive input from different environments [Shimizu, 2012]. LiNGAM is a method also developed for causal discovery, but relies on a rather different set of assumptions: LiNGAM does not assume heterogeneity of the environments, but instead requires non-Gaussian noise variables for parent identification. On the other hand, L-ICP relies on environmental heterogeneity. We contrast the two methods by showcasing their performance on a spectrum of settings spanning both assumptions. In particular we generate data once with uniform noise, matching LiNGAMs assumptions, and once with Gaussian noise, matching L-ICPs assumptions. We introduce heterogeneity into the data by dividing $E = 30$ environments into two groups that have inter-group heterogeneity but intra-group homogeneity, as this allows for a controlled way of inducing heterogeneity. The parameter $I_S$ from Theorem 2 provides a natural way to quantify the heterogeneity in the various scenarios and we define $h := \max_{S:S^* \not\subseteq S} I_S$ as the heterogeneity parameter.[2] Given the exponential relationship of the Theorem, we report in Figures 3 and 4 the performance of both methods along the parametrization $-\ln(h) \in [0, \infty)$, so that larger values indi-

---

[2]Note that while the parameter $h$ is still meaningful, the other data generation assumptions made by Theorem 2 do not hold.

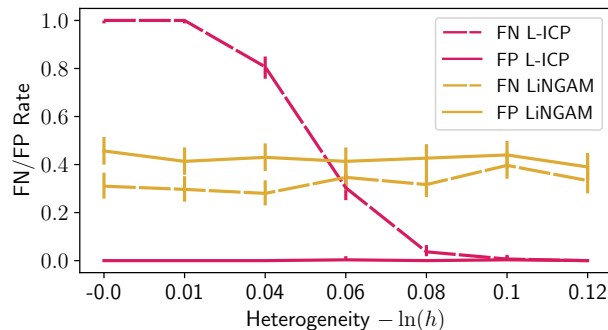

Figure 3: A comparison of joint LiNGAM and L-ICP under uniform noise.

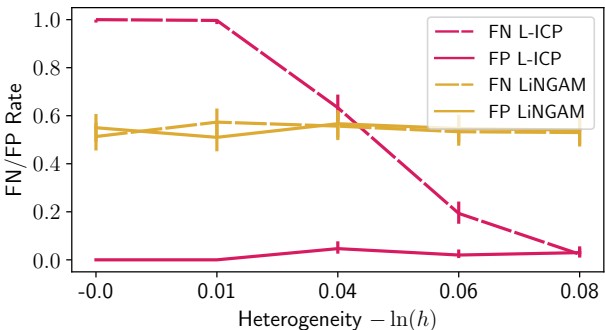

Figure 4: A comparison of joint LiNGAM and L-ICP under Gaussian noise.

cate stronger heterogeneity and $-\ln(h) = 0$ indicates that no heterogeneity was present. As expected, without environment heterogeneity L-ICP fails to identify causal parents, while adding a moderate amount of heterogeneity eventually leads to near-optimal performance. LiNGAM is largely unaffected by changes of the heterogeneity, while its overall performance, even in the well-specified case of the uniform noise, is quite low: note that an algorithm that in each run alternates between reporting the empty set and all covariates would achieve a false positive and false negative rate of $\frac{1}{2}$, as we only count if *a* false positive/negative was present. In Section B.1 of the Appendix we perform the same type of experiment with a scaled Student-t distribution. The results of that experiment show that for a low degree of freedom of 3, and thus a strong Gaussanity violation, L-ICP shows bad performance and also increased heterogeneity does not help in recovering a good performance. For a moderate degree of freedom of 10 stronger heterogeneity does help again.

## 6.1 NETWORK DETECTION IN DYNAMICAL SYSTEMS

Finally, we want to describe, and showcase, that one may use L-ICP for network detection in dynamical systems. Following the remarks after Theorem 1, finding a full causal

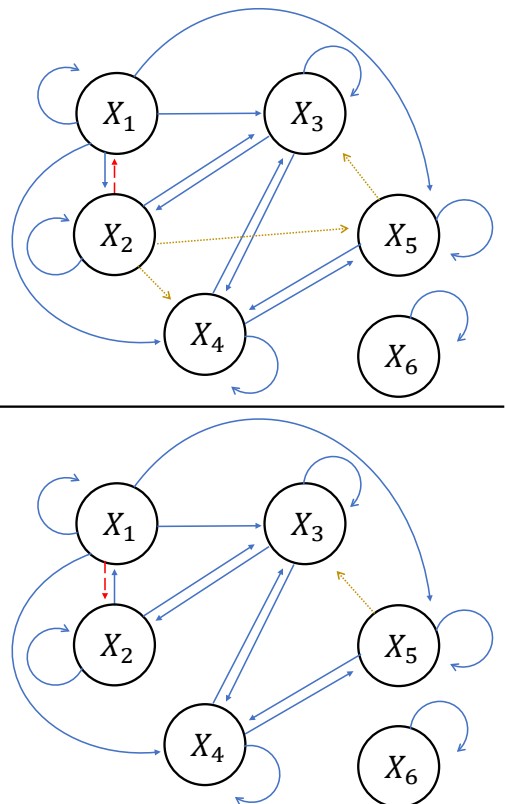

Figure 5: The reported causal graphs.
**Top:** L-ICP, **Bottom:** PCMCI
**Solid blue:** Correctly found. **Dashed red:** Not found (false negative discovery). **Dotted yellow:** Falsely reported (false positive discovery.)

graph is possible in our proposed setting if no covariate is directly affected by the environment index, but only indirectly through changing structural parameters. In this section we simulate data from a non-linear dynamical system, which approximately follows this setting when we consider our local models as locally linear approximations of the system. In this experiment we want to reveal if the non-linearity of the system can introduce sufficient heterogeneity across time so that L-ICP can subsequently discover causal relations. More precisely, our data consists of a discrete-time and noisy version of a five-dimensional Lorenz system described by Shen [2014] together with an independently sampled random walk. The precise equations of the system can be found in Section B.2 in the supplementary material.

Given that we have dynamical data, we chose our environments to be time intervals of length $n$. More precisely, for a given starting time $t_0 \in \mathbb{N}$ we define the observations in environment $e_{t_0}$ by $(X^{e_{t_0}}, Y^{e_{t_0}}) := \{(X^t, Y^t)\}_{t_0 \leq t \leq t_0 + n}$. The target variable $Y^t$ is now the observation of *any* chosen covariate, but at the next time-step. For example, if we want to find the causal parents of $X_1$, we define $Y^t := X_1^{t+1}$.

**Experimental details.** First, over 500 independent runs we generated 8500 samples of the dynamical system. Given the data of one run, we split the time series into intervals of length $n$ and then run L-ICP with $B = 500$ and $\alpha = 0.1$ using those intervals as environments. For that, we need to decide the interval length $n$, which we did with the following rationale: If $n$ is chosen very small, the algorithm tends to return the empty set because most subsets $S \subseteq [D]$ are plausible causal parents. If $n$ is too large, the method tends also to return the empty set, as in that case no subset of covariates provides a set of plausible causal parents due to a strong violation of the linearity assumption. We thus first tested for which sample sizes $n$ the method tends to *not* return the empty set for any covariate in an individual run. Leaving the first 500 samples as a warm-up phase for the system, and splitting the remaining data into $E = 300$ intervals of length $n$, the method tended to return a non-empty output for $n \in [15, 35]$. This motivated our choice to set $n = 25$. Over the 500 runs we then count for each target how often each covariate was reported as a causal parent. While the full counts are found in Section B.2, we now report a causal graph with the following reasoning: given that we run L-ICP with $\alpha = 0.1$ we consider all covariates that were reported in significantly over 10% of the runs as causal parents as true causal parents. The significance is tested with a binomial test at significance level 0.05, with the null that a covariate is found in 10% of the runs, and the alternative that the reported rate is greater than 10%. We compare this approach with PCMCI [Runge et al., 2019], a causal discovery method targeting time-series data. PCMCI first uses conditional independence tests on lagged variables to find a graph skeleton, and then orients the edges along their temporal direction. For a fair comparison we use PCMCI with a partial correlation test, matching our linearity assumption, and in each run we perform PCMCI with a target confidence level of 0.1 on 30 evenly spaced intervals of length 25. While in each run we have access to a total of 300 intervals, we only perform PCMCI on 30 of those for computational reasons. We then count how often each arrow was in total reported. Reporting the final results with the same reasoning as for L-ICP resulted in a fully connected graph, so we instead picked a threshold that is tuned based on the ground truth graph.

In the top of Figure 5 we report the causal graph computed with L-ICP and we can affirm that non-linear relationships in a dynamical system can introduce sufficient heterogeneity across time. We find all but one connection, while we reported three incorrect edges. By Proposition 2 we know that this has to be due to model misspecification, highlighting an important limitation of the linearity assumption. In comparison PCMCI (bottom of Figure 5) also found all but one connection, but only reported one incorrect edge. Given that we clairvoyantly thresholded the results of PCMCI and, in contrast to L-ICP, PCMCI can not deal with instantaneous relationships, one may consider L-ICP competitive.

# 7 DISCUSSION

In this paper we presented an extension of the work from Peters et al. [2016] to a setting where models are estimated locally in every environment, with many interesting consequences. We now discuss limitations, how they can be addressed, and possible extensions of our work.

**Scalability of the method.** The computational cost of our current proposal scales exponentially with respect to the dimension $D$. To overcome this bottleneck, one could, instead of looping over all possible subsets of $[D]$, greedily add or remove covariates as plausible causal parents. While the greedy removal was already applied to ICP by Salas-Porras et al. [2022], it is clear that such methodology can generally not enjoy the same guarantees as the full method, but an extensive comparison against the full method is an interesting open task. Alternatively, one can reduce the dimensionality $[D]$, for example by clustering highly correlated variables. Instead of looping over all subsets from $[D]$, one may then loop over all clusters, and the method reports clusters in which a parent is present. Finally, we envision a procedure where the test-criterion given by $\phi_S$ is encoded as an objective function that one may optimize over the set of covariates, in the spirit of procedures such as the LASSO. Similar ideas have been applied in the machine learning literature, see for example Arjovsky et al. [2019].

**Extension to non-Gaussian noise.** To calibrate our hypothesis test, we assume that the noise is normally distributed. While this provides a starting point to analyze a specific methodology within our proposed framework (L-ICP), we observe in Figures 8a and 8b that a violation of this assumption can lead to a strong performance loss. The normality assumption, however, is *not* an integral part of the methodology, and Algorithm 1 can be run by replacing our hypothesis test $\phi_S(\boldsymbol{X}, \boldsymbol{Y}, \alpha)$ with a different one that relies on other assumptions. We could instead calibrate the test by permutation as done by Stoepker et al. [2022], e.g., by permuting the residuals over all environments and contrasting the test statistic in the permuted and unpermuted data. Alternatively, one may use Levene's test for equal variances [Brown and Forsythe, 1974], which has robustness against non-normality and was used in a similar context by Heinze-Deml et al. [2017]. Finally note that the distribution of the sum of squared residuals $\|r^e_{S*}\|^2_2$ will be approximately normal when $n^e$ is large, as a consequence of the central limit theorem, which one might be able to capitalize on with modifications to our test statistic.

**The role of locality.** The main novelty of our proposed setting is that we model each environment separately with a local model without any additional structural assumptions between the local models. This relaxes the global linearity assumption and, in some sense, allows us to model non-

linear systems as seen in Section 6.1. But more importantly, in some scenarios the data generation in different environments can truly follow different functional relationships, and a local model becomes necessary. Thinking about different car types as different environments, it is reasonable to assume that the causal relations are the same, but the structural equations are not.

**Limitations of linearity.** In the experiments of Section 6.1 we approximated a non-linear dynamical systems with intervals of linear models. To ensure approximate linearity we had to set the length of the intervals, which corresponds to the sample size $n$ of the environments within our setting, at a relatively small number of $n = 25$. In that experiment this sample size was too small to recover all causal parents, while some false discoveries are likely due the violation of the linearity assumption. While both problems highlight the need for non-linear versions of the proposed methodology, the general concept of (L-)ICP does not hinge on linearity, as also pointed out in Peters et al. [2016].

**Changing distribution of the target noise.** Arguably one of the strongest assumptions in our setting is that the target noise $\varepsilon^e$ follows the same distribution in every environment. While this ensures that the environment cannot have any influence on $\varepsilon^e$, this assumption can be relaxed. We may, for example, parameterize the distribution of $\varepsilon^e$ along a parameter $\theta$ and then assume that $\theta$ and the environment indices are independently sampled from a distribution. With that assumption, the environment index is independent of $\theta$ and thus should not have any effect on the distribution of $\varepsilon^e$. Establishing that a set $S \subseteq [D]$ is plausible then can, for example, be established by testing the independence of the environment index and $\theta$.

**Picking good environments.** In applications we may often face the choice of how to define the different environments in which our data is partitioned. In the experiment of Section 6.1, for example, we have access to a stream of dynamical data and need to split this stream in a meaningful way. While in this experiment we simply picked environments that are evenly spaced in time, one can think about more sophisticated ways of picking environments and identifying good environments should be a focus of future research. This is not only important for our own methodology, but any method that views an environment as an accidental interventions.

### Acknowledgements

This research has been funded by NWO under the grant PrimaVera (https://primavera-project.com) number NWA.1160.18.238 and was done in collaboration with our project partner ASML.

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

# Invariant Causal Prediction with Local Models
## (Supplementary Material)

**Alexander Mey**[1,2]  **Rui M. Castro**[1,3]

[1]Department of Mathematics and Computer Science, Eindhoven University of Technology, Eindhoven, The Netherlands
[2]ASML Research, Veldhoven, The Netherlands
[3]Eindhoven Artificial Intelligence Systems Institute (EAISI), Eindhoven University of Technology, Eindhoven, The Netherlands

## A  PROOFS

### A.1  PROOF OF LEMMA 1

**Lemma 1**  *If $\tilde{H}_{0,S}$ is true, then so is $H_{0,S}$.*

*Proof.* If $\tilde{H}_{0,S}$ is true then, using the independence of the residual noise $r^e$ and covariates, we know that $\gamma^e$ as well as $\tilde{\beta}_S^e$ are solutions of the least squares problem

$$\min_\beta \mathbf{E}\left[\sum_{i=1}^{n^e}(X_{S,i}^e\beta - Y_i^e)^2\right].$$

This means that $X_{S,i}^e\tilde{\beta} = X_{S,i}^e\gamma^e$ almost surely, which implies the lemma. □

### A.2  PROOF OF PROPOSITION 2

*Proof.* If for all $e \in [E]$ the matrix $(X_{S^*}^e)^T X_{S^*}^e$ is not invertible we are done, as in that case $\tilde{S} = \emptyset$. If this matrix is invertible, then we know by Assumptions 1 and 3 that $r_{S^*}^e \sim \sigma_Y^2\chi^2(n^e - \text{rank}((X_{S^*}^e)^T X_{S^*}^e))$. With that $T_{S^*} = \min_{e\in[E]}\|r_{S^*}^e\|_2^2/\max_{e\in[E]}\|r_{S^*}^e\|_2^2$ and $\min_{e\in[E]}Z_S^e/\max_{e\in[E]}Z_S^e$ follow the same distribution by definition of $Z_S^e$, which implies that

$$\mathbf{P}\left(\mathbf{P}(T_{S^*}(\boldsymbol{X},\boldsymbol{Y}) > \frac{\min_{e\in[E]}Z_S^e}{\max_{e\in[E]}Z_S^e} \mid \boldsymbol{X},\boldsymbol{Y}) \le \alpha\right) \le \alpha.$$

By definition of $\phi_{S^*}$ this means that $\mathbf{P}(\phi_{S^*} = 1) \le \alpha$, which by definition of $\tilde{S}$ finally implies that $\mathbf{P}(\tilde{S} \not\subseteq S) \le \alpha$. □

### A.3  PROOF OF THEOREM 1

*Proof.* The general proof strategy is the following: From the two distinct environments $e, v \in [E]$ we pick a sample $i \in [n^e]$ and $j \in [n^v]$. If the hypothesis $H_{0,S}$ would be true we may conclude that the population residuals $r_i^e$ and $r_j^v$ have the same distribution, and therefore the same variance. We then show, however, that there exists a $u \in S^*$ and $u_1 \in [D]$ such that $\mathbf{V}[r_i^e]$ is a proper rational function (so the ratio of two polynomials) of finite degree with respect to $\beta_u^e$ for almost all choices of $\Delta_{u_1,i}^e$. Fixing all entries of $(\beta^e, \beta^v, \Delta^e)$ except $\beta_u^e$ and $\Delta_{u_1,i}^e$ at arbitrary values, we conclude that the equation $\mathbf{V}[r_i^e] = \mathbf{V}[r_j^v]$ can be solved for at most finitely many values of $\beta_u^e$ and $\Delta_{u_1,i}^e$. This means that $M_0$, the solution space of the equation $\mathbf{V}[r_i^e] = \mathbf{V}[r_j^v]$ with respect to $(\beta^e, \beta^v, \Delta^e)$, has Lebesgue measure zero. This finally implies that the hypothesis $H_{0,S}$ is false for all parameter choices outside of $M_0$.

*Accepted for the 40th Conference on Uncertainty in Artificial Intelligence* (UAI 2024).

To start the proof, we note that $\mathbf{V}[r_i^e]$ is always a rational function of polynomials of finite degree with respect to the entries of $\beta^e$. On the one hand this follows from Assumption 2, ensuring that every covariate $d$ with $y \in \mathbf{AN}(d)$ is a polynomial in $Y^e$ and thus a polynomial in $\beta^e$. On the other hand we know from Constales [1998] that the Moore-Penrose inverse has a closed form solution consisting only of elementary operations. The rest of the proof is concerned with showing that there is at least one $u \in S^*$ such that $\mathbf{V}[r_i^e]$ is a proper polynomial with respect to $\beta_u^e$, meaning that the leading term in $\beta_u^e$ does not vanish. To simplify notation, we define

$$P(S, S^*) := \mathbf{E}[(X_S^e)^t X_S^e)]^\dagger \mathbf{E}[(X_S^e)^t X_{S^*}^e] \in \mathbb{R}^{|S| \times |S^*|}$$

and

$$P(\varepsilon^e) := \mathbf{E}[(X_S^e)^t X_S^e)]^\dagger \mathbf{E}[(X_S^e)^t \varepsilon^e] \in \mathbb{R}^{|S|}$$

and note that $\tilde{\beta}_S^e = P(S, S^*)\beta_{S^*}^e + P(\varepsilon^e)$. We split the proof into two cases.

**The first case** assumes that no variable in $S$ is a descendant of $Y$. In that case we pick any $u \in S^* \setminus S$ and split the variance of the residual as follows:

$$\mathbf{V}[r_i^e]$$
$$= \mathbf{V}[X_{S^*,i}^e \beta_{S^*}^e + \varepsilon_i^e - X_{S,i}^e \tilde{\beta}_S^e] = \mathbf{V}[(X_{S^*,i}^e - X_{S,i}^e P(S, S^*))\beta_{S^*}^e + \varepsilon_i^e - X_{S,i}^e P(\varepsilon^e)]$$
$$= \mathbf{V}\Bigg[(X_{u,i}^e - X_{S,i}^e P(S, S^*)_{\cdot,u})\beta_u^e$$

$$+ \sum_{d \in S^* \setminus \{u\}} (X_{d,i}^e - X_{S,i}^e P(S, S^*)_{\cdot,d})\beta_d^e + \varepsilon_i^e - X_{S,i}^e P(\varepsilon^e)\Bigg]$$
$$= (\beta_u^e)^2 \mathbf{V}[(X_{u,i}^e - X_{S,i}^e P(S, S^*)_{\cdot,u})] \tag{9}$$

$$+ \mathbf{V}\Bigg[\sum_{d \in S^* \setminus \{u\}} (X_{d,i}^e - X_{S,i}^e P(S, S^*)_{\cdot,d})\beta_d^e + \varepsilon_i^e - X_{S,i}^e P(\varepsilon^e)\Bigg] \tag{10}$$

$$+ 2\beta_u^e \mathbf{C}\Bigg[(X_{u,i}^e - X_{S,i}^e P(S, S^*)_{\cdot,u}), \sum_{d \in S^* \setminus \{u\}} (X_{d,i}^e - X_{S,i}^e P(S, S^*)_{\cdot,d})\beta_d^e + \varepsilon_i^e - X_{S,i}^e P(\varepsilon^e)\Bigg]. \tag{11}$$

We now argue that the variance in line (9) does not vanish (**Claim 1** further below), and that the covariance in line (11) is only linear in $\beta_u^e$ (**Claim 2** further below). With that, and noting that the variance term in line (10) is always positive, we know that there are constants $a > 0$ and $b \in \mathbb{R}$ (which depend on the distributions of the covariates) such that

$$\mathbf{V}[r_i^e] \geq (\beta_u^e)^2 a + \beta_u^e b. \tag{12}$$

With that $\mathbf{V}[r_i^e]$ is a proper polynomial in $\beta_u^e$, as $\mathbf{V}[r_i^e] \to \infty$ for $\beta_u^e \to \infty$.

**Claim 1.** The variance in line (9) does not vanish. Proof: Let $\bar{S} \subset S$ be the set of all indices $d \in S$ with $P(S, S^*)_{d,u} \neq 0$. All indices outside $\bar{S} \cup \{u\}$ are irrelevant as the corresponding covariate vanishes within the variance term (9). Now let $u_0$ be a sink node in $\bar{S} \cup \{u\}$, which implies by Assumption 2 that $\Delta_{u_0,i}^e$ is independent of all other covariates with index in $\bar{S} \cup \{u\}$. First, let $u_0 \in \bar{S}$, then we may split the variance from line (9) as

$$\mathbf{V}[(X_{u,i}^e - X_{S,i}^e P(S, S^*)_{\cdot,u})]$$
$$= \mathbf{V}[X_{u,i}^e - X_{S,i}^e P(S, S^*)_{\cdot,u} + \delta_{u_0,i}^e P(S, S^*)_{u_0,u}] + \mathbf{V}[\delta_{u_0,i}^e]P(S, S^*)_{u_0,u}^2.$$

This splitting is allowed as by design $\delta_{u_0,i}^e$ is independent of all covariates $X_{d,i}^e$ within the variance with $d \neq u_0$, and the dependence on $X_{u_0,i}^e$ is canceled by the term $+\delta_{u_0,i}^e P(S, S^*)_{u_0,u}$. From the definition above we know that $P(S, S^*)_{u_0,u} \neq 0$ and by Assumption 2 we have $\mathbf{V}[\delta_{u_0,i}^e] > 0$, which implies **Claim 1**. The case that $u_0 = u$ follows analogously by splitting $\delta_{u,i}^e$ out of the variance.

**Claim 2.** The covariance in line (11) is only linear in $\beta_u^e$. Proof: This is a direct consequence from the fact that no covariate is a descendant of $Y^e$, which also implies that no covariate has a dependence on $\beta_u^e$. The covariance does then not depend on $\beta_u^e$, and we only have a linear dependency in line (11) from the leading coefficient.

**The second case** assumes there exists at least one $d \in S$ with $d \in \mathbf{DE}(y)$, where by acyclicity of $G$ we know that $d \notin S^*$. Let $u \in S^* \setminus S$ and we fix all entries of $(\beta^e, \beta^v)$, except $\beta_u^e$. We now show that there is an index $u_1 \in [D]$ such that for almost all values of $\Delta_{u_1,i}^e$ the variance term $\mathbf{V}[r_i^e]$ is different for $\beta_u^e = 0$ and $\beta_u^e \to \infty$. This implies that $\mathbf{V}[r_i^e]$ is for almost all values of $\Delta_{u_1,i}^e$ a ratio of proper polynomial in $\beta_u^e$ and the argumentation follows as in the first case. To make this argumentation formal we use the following two claims:

**Claim 3.** For $\beta_u^e = 0$ and any $u_0 \in \mathbf{DE}(y)$ the term $\mathbf{V}[r_i^e]$ is finite for $\Delta_{u_0,i}^e \to \infty$. Proof: We just have to note that $u_0 \notin S^*$, which implies that $\mathbf{V}[Y_i^e]$ remains finite for $\Delta_{u_0,i}^e \to \infty$. The claim follows by noting that $\mathbf{V}[r_i^e] \leq \mathbf{V}[Y_i^e] + \mathbf{E}[Y_i^e]^2$.

**Claim 4.** For all $u_0 \in \mathbf{DE}(y)$ set $\Delta_{u_0,i}^e = q$, then for $q \to \infty$ the term $\mathbf{V}[r_i^e]$ diverges for $\beta_u^e \to \infty$. Proof: We may assume that there exists at least one $u_0 \in \mathbf{DE}(y)$ such that $\tilde{\beta}_{u_0}^e \neq 0$ for $q \to \infty$, otherwise we are back to the first case as we then may remove all descendants of $y$ from our set $S$. In this first case we have shown **Claim 4** already after Inequality (12). Without loss of generality let $u_0 \in \mathbf{DE}(y)$ be a sink node, which implies that $\delta_{u_0}$ is independent of all other variables by (5). Then we may split $\mathbf{V}[r_i^e]$ as

$$\mathbf{V}[r_i^e] = \mathbf{V}[Y - X_S^e \tilde{\beta}_S^e + \delta_{u_0,i}^e \tilde{\beta}_{u_0}^e - \delta_{u_0,i}^e \tilde{\beta}_{u_0}^e]$$
$$= \mathbf{V}[Y - X_S^e \tilde{\beta}_S^e + \delta_{u_0,i}^e \tilde{\beta}_{u_0}^e] + \mathbf{V}[\delta_{u_0,i}^e](\tilde{\beta}_{u_0}^e)^2.$$

Since by assumption $\tilde{\beta}_{u_0}^e \neq 0$ we observe that $\mathbf{V}[r_i^e] \to \infty$ for $\mathbf{V}[\delta_{u_0,i}^e] = \Delta_{u_0,i}^e \to \infty$.

The above **Claim 3** and **Claim 4** together imply that $\mathbf{V}[r_i^e]$ obtain different values for $\beta_u^e = 0$ and $\beta_u^e \to \infty$ in the regime that $\Delta_{u_0,i}^e \to \infty$ for all $u_0 \in \mathbf{DE}(y)$. This implies that in this regime the term $\mathbf{V}[r_i^e]$ is a proper rational function in $\beta_u^e$. It could, however, still happen that for specific choices of $\Delta_{\cdot,i}^e \in (0, \infty)^D$ the rational function dependence on $\beta_u^e$ cancels within $\mathbf{V}[r_i^e]$. More precisely, let $c$ be the leading coefficient of the polynomial term in $\beta_u^e$, then it is still possible that $c = 0$ for specific choices of $\Delta_{\cdot,i}^e$ as $c$ generally depends on those terms. However, let $u_1 \in [D]$ be any index such that $c$ depends on $\Delta_{u_1,i}^e$. Fixing all entries in $\Delta_{\cdot,i}^e$ except $\Delta_{u_1,i}^e$ we know that $c = 0$ for at most finitely many choices of $\Delta_{u_1,i}^e$ since $\mathbf{V}[r_i^e]$ is also a rational function in $\Delta_{u_1,i}^e$.

We thus have shown that there exists a $u \in S^* \setminus S$ such that $\mathbf{V}[r_i^e]$ is a proper rational function in $\beta_u^e$ for almost all values of $\Delta_{\cdot,i}^e$. With that we know that for almost all values of $\Delta_{\cdot,i}^e$ the equation $\mathbf{V}[r_i^e] = \mathbf{V}[r_i^v]$ can be solved for at most finitely many choices of $\beta_u^e$. With that, the solution space $M_0$ of the equation $\mathbf{V}[r_i^e] = \mathbf{V}[r_i^v]$ with respect to the parameters $(\beta^e, \beta^v, \Delta^e)$ has a Lebesgue measure of zero. As the equality of $\mathbf{V}[r_i^e]$ and $\mathbf{V}[r_i^v]$ is a necessary condition for $H_{0,S}$ to be true, we know that $H_{0,S}$ is false for all parameter values outside of $M_0$.

$\square$

## A.4 PROOF OF THEOREM 2

We first collect some lemmas that are needed for the main proof.

**Lemma 2.** *For any two random variables $X, Y$ and $c > 0$ it holds that*

$$\mathbf{P}(X < Y) \leq 2(\mathbf{P}(X < c) + \mathbf{P}(Y > c)).$$

*Proof.* First we define the three events

$$A = \{(X < c \wedge Y < c) \wedge (X < Y)\}$$
$$B = \{(X > c \wedge Y > c) \wedge (X < Y))\}$$
$$C = \{X < c \wedge Y > c\}.$$

With that we may conclude that

$$
\begin{aligned}
\mathbf{P}(X < Y) = \mathbf{P}(A \vee B \vee C) &= \mathbf{P}(A) + \mathbf{P}(B) + \mathbf{P}(C) \\
&\leq \mathbf{P}(X < c) + \mathbf{P}(B) + \mathbf{P}(C) \\
&\leq \mathbf{P}(X < c) + \mathbf{P}(Y > c) + \mathbf{P}(X < c \wedge Y > c) \\
&= \mathbf{P}(X < c) + \mathbf{P}(Y > c) + \mathbf{P}(X < c) + \mathbf{P}(Y > c) - \mathbf{P}(X < c \vee Y > c) \\
&\leq 2(\mathbf{P}(X < c) + \mathbf{P}(Y > c)).
\end{aligned}
$$

$\square$

**Lemma 3** (Adapted from Dasgupta and Gupta [2003], Lemma 2.2). *Let $Z \sim \chi^2(k)$ and $F_Z$ be the cumulative distribution function of $Z$. Then then following two inequalities hold:*

$$1 - F_Z(kz) \leq (z)^{\frac{k}{2}} e^{\frac{k}{2}(1-z)} \text{ for } z > 1 \tag{13}$$

$$F_Z(kz) \leq (z)^{\frac{k}{2}} e^{\frac{k}{2}(1-z)} \text{ for } 0 < z < 1 . \tag{14}$$

*Proof.* We begin with the second inequality. For $1 \leq i \leq k$ let $X_i \sim \mathcal{N}(0,1)$ be $k$ independent standard normal random variables. Then $Z := \sum_{i=1}^{k} X_i^2 \sim \chi^2(k)$. We use a Chernoff bounding technique as follows and for $t > 0$ we derive

$$
\begin{aligned}
F_Z(kz) = \mathbf{P}(kz - \sum_{i=1}^{k} X_i^2 \geq 0) &= \mathbf{P}(e^{tkz - t\sum_{i=1}^{k} X_i^2} \geq 1) \\
&\leq e^{tkz} \mathbf{E}\left[ e^{-t\sum_{i=1}^{k} X_i^2} \right] = e^{tkz} \mathbf{E}\left[ e^{-tX_i^2} \right]^k .
\end{aligned}
$$

Using the known equality $\mathbf{E}\left[ e^{-tX_i^2} \right] = (1 + 2t)^{-\frac{1}{2}}$ for $-\frac{1}{2} < t < \infty$ we may set $t = \frac{1}{2}\frac{1-z}{z}$ since $z < 1$ to obtain

$$e^{tkz} \mathbf{E}\left[ e^{-tX_i^2} \right]^k = e^{tkz}(1 + 2t)^{-\frac{k}{2}} = e^{\frac{k}{2}(1-z)} z^{\frac{k}{2}} .$$

The first inequality of the lemma follows similarly. $\square$

**Lemma 4.** *Let $S \subset [D]$ with $S^* \not\subseteq S$ and set $U := S \setminus S^*$. Then, under Assumptions 1, 3 and 4 we have the following properties of the SSR when regressing $Y$ onto $X_S$ in environments $v \in [E_1]$ and $w \in [E_2]$. Defining $\rho^v := \sum_{u \in U} (\beta_u^1)^2 (\sigma_u^v)^2 + (\sigma_Y)^2$ and $\rho^w := \sum_{u \in U} (\beta_u^2)^2 (\sigma_u^w)^2 + (\sigma_Y)^2$ it holds that $\frac{1}{\rho^v}\|r_S^v\|_2 \sim \chi^2(n - |S|)$ and $\frac{1}{\rho^w}\|r_S^w\|_2 \sim \chi^2(n - |S|)$. Here $r_S^v$ and $r_S^w$ are the residuals defined in Algorithm 1.*

*Proof.* Regressing the target $Y$ only on the covariates in $S$ we obtain in environment $v \in [E_1]$ the linear model

$$Y^v = \beta_S^1 X_S^v + r_S^v,$$

where $r_S^v = \beta_U^1 X_U^v + \varepsilon^e$. Because of the normality and independence assumptions, we find that $r_S^v$ follows a normal distribution with variance $\rho^v = \sum_{u \in U} (\beta_u^1)^2 (\sigma_u^v)^2 + (\sigma_Y)^2$. Furthermore $r_S^v$ can be assumed to be of zero mean, due to our inclusion of a column of constant ones in $X_S^v$. This implies that $P^v := \frac{1}{\rho^v}\|r_S^v\|_2 \sim \chi^2(k)$ and we can define the equivalent expression for $w \in [E_2]$ by setting $Q^w := \frac{1}{\rho^w}\|r_S^w\|_2 \sim \chi^2(k)$. $\square$

**Proof of Theorem 2.**

*Proof.* To identify when our method accepts $S$ as a set of potential causal parents, we need to control the probability $\mathbf{P}(\phi_S = 0)$. For convenience, we introduce $Z_{\min} := \min_{e \in [E]} Z_S^e$ and $Z_{\max} := \max_{e \in [E]} Z_S^e$. The probability of a false negative can be bounded by Markov's Inequality as

$$\mathbf{P}(\phi_S(\boldsymbol{X}, \boldsymbol{Y}) = 0) = \mathbf{P}\left( \mathbf{P}\left( T(\boldsymbol{X}, \boldsymbol{Y}) > \frac{Z_{\min}}{Z_{\max}} \,\bigg|\, \boldsymbol{X}, \boldsymbol{Y} \right) \geq \alpha \right) \tag{15}$$

$$\leq \frac{1}{\alpha} \mathbf{P}\left( T(\boldsymbol{X}, \boldsymbol{Y}) > \frac{Z_{\min}}{Z_{\max}} \right) . \tag{16}$$

We continue with analyzing the quantity $\mathbf{P}(T(\boldsymbol{X},\boldsymbol{Y}) > Z_{\min}/Z_{\max})$ and in particular try to understand the distribution of $T(\boldsymbol{X},\boldsymbol{Y})$ under Assumption 4. By Lemma 4 we know that $P^v := \frac{1}{\rho^v}\|r_S^v\|_2 \sim \chi^2(n-|S|)$ and $Q^w := \frac{1}{\rho^w}\|r_S^w\|_2 \sim \chi^2(n-|S|)$ for $\rho^v := \sum_{u\in U}(\beta_u^1)^2(\sigma_u^v)^2 + (\sigma_Y)^2$ and $\rho^w := \sum_{u\in U}(\beta_u^2)^2(\sigma_u^w)^2 + (\sigma_Y)^2$. This allows us to write:

$$T(\boldsymbol{X},\boldsymbol{Y}) = \frac{\min_{v\in[E_1],w\in[E_2]}(\min(\rho^v P^v, \rho^w Q^w))}{\max_{v\in[E_1],w\in[E_2]}(\max(\rho^v P^v, \rho^w Q^w))}.$$

With the help of Lemma 2 we may then for any $c > 0$ bound

$$\mathbf{P}\left(T(\boldsymbol{X},\boldsymbol{Y}) > \frac{Z_{\min}}{Z_{\max}}\right) \leq \mathbf{P}\left(\frac{\min_{w\in[E_2]}\rho^w Q^w}{\max_{v\in[E_1]}\rho^v P^v} > \frac{Z_{\min}}{Z_{\max}}\right)$$

$$\leq 2\mathbf{P}\left(\frac{\min_{w\in[E_2]}\rho^w Q^w}{\max_{v\in[E_1]}\rho^v P^v} > c\right) + 2\mathbf{P}\left(c > \frac{Z_{\min}}{Z_{\max}}\right). \tag{17}$$

With the above inequality we are allowed to continue with the expression in line (17) and we start with the first term $\mathbf{P}(\min_{w\in[E_2]}\rho^w Q^w / \max_{v\in[E_1]}\rho^v P^v > c)$. By setting $c = \sqrt{\frac{\rho^w}{\rho^v}} < 1$ we have that $c\frac{\rho^v}{\rho^w} = \frac{1}{c}$, which allows us to write

$$\mathbf{P}\left(\frac{\min_{w\in[E_2]}Q^w}{\max_{v\in[E_1]}P^v} > c\frac{\rho^v}{\rho^w}\right) = \mathbf{P}\left(\frac{\max_{v\in[E_1]}P^v}{\min_{w\in[E_2]}Q^w} < c\right),$$

which is now the quantity we study further. We further want to simplify this term by splitting it with the help of the two events

$$\mathcal{E}_1 := \left\{\frac{\max_{v\in[E_1]}P^v}{\min_{w\in[E_2]}Q^w} < c\right\}$$

$$\mathcal{E}_2 := \left\{\max_{v\in[E_1]}P^v > k\sqrt{c} \wedge \min_{w\in[E_2]}Q^w < k\frac{1}{\sqrt{c}}\right\}.$$

Noting that $\mathbf{P}(\mathcal{E}_1, \mathcal{E}_2) = 0$ we can bound

$$\mathbf{P}\left(\frac{\min_{w\in[E_2]}Q^w}{\max_{v\in[E_1]}P^v} > c\frac{\rho^v}{\rho^w}\right) = \mathbf{P}\left(\frac{\max_{v\in[E_1]}P^v}{\min_{w\in[E_2]}Q^w} < c\right) = \mathbf{P}(\mathcal{E}_1)$$

$$\leq \mathbf{P}(\mathcal{E}_1, \mathcal{E}_2) + \mathbf{P}(\{\mathcal{E}_2\}^c) \tag{18}$$

$$\leq \mathbf{P}\left(\max_{v\in[E_1]}P^v < k\sqrt{c}\right) + \mathbf{P}\left(\min_{w\in[E_2]}Q^w > k\frac{1}{\sqrt{c}}\right). \tag{19}$$

With the reminder that $c < 1$ and that for any $v \in [E_1]$ and $w \in [E_2]$ the terms $P^v$ and $Q^w$ follow a Chi-square distribution with $k$ degrees of freedom we can use Lemma 3 to conclude that for any $v_0 \in [E_1]$

$$\mathbf{P}\left(\max_{v\in[E_1]}P^v < k\sqrt{c}\right) \leq \mathbf{P}(P^{v_0} < k\sqrt{c}) \leq (\sqrt{c})^{\frac{k}{2}}e^{\frac{k}{2}(1-\sqrt{c})}. \tag{20}$$

And similarly, we derive

$$\mathbf{P}\left(\min_{w\in[E_2]}Q^w > k\frac{1}{\sqrt{c}}\right) \leq \left(\frac{1}{\sqrt{c}}\right)^{\frac{k}{2}}e^{\frac{k}{2}\left(1-\frac{1}{\sqrt{c}}\right)}. \tag{21}$$

With this we can control the first term of our intermediate target defined in (17) as plugging Inequalities (20) and (21) into (19) provides the result

$$\mathbf{P}\left(\frac{\max_{v\in[E_1]}P^v}{\min_{w\in[E_2]}Q^w} < c\right) \leq (\sqrt{c})^{\frac{k}{2}}e^{\frac{k}{2}(1-\sqrt{c})} + \left(\frac{1}{\sqrt{c}}\right)^{\frac{k}{2}}e^{\frac{k}{2}\left(1-\frac{1}{\sqrt{c}}\right)}. \tag{22}$$

The other term in our target (17) is given by $\mathbf{P}(Z_{\min}/Z_{\max} < c)$. To bound this term, we can use almost the exact same reasoning as for the first term, the only difference being the dependence on $E$ as for this term we use a union bound in the inequalities that correspond to (20) and (21) for the previous term. In the end we obtain that

$$\mathbf{P}\left(\frac{Z_{\min}}{Z_{\max}} < c\right) \leq E\left((\sqrt{c})^{\frac{k}{2}} e^{\frac{k}{2}(1-\sqrt{c})} + \left(\frac{1}{\sqrt{c}}\right)^{\frac{k}{2}} e^{\frac{k}{2}\left(1-\frac{1}{\sqrt{c}}\right)}\right). \tag{23}$$

Plugging the result from (22) and (23) back into (17) we obtain

$$\mathbf{P}\left(T(\boldsymbol{X},\boldsymbol{Y}) > \frac{Z_{\min}}{Z_{\max}}\right) \leq 4E\left((\sqrt{c})^{\frac{k}{2}} e^{\frac{k}{2}(1-\sqrt{c})} + \left(\frac{1}{\sqrt{c}}\right)^{\frac{k}{2}} e^{\frac{k}{2}\left(1-\frac{1}{\sqrt{c}}\right)}\right).$$

Finally, plugging this back into Inequality (16) and noting that $c = \sqrt{\frac{\rho^w}{\rho^v}}$, we obtain the statement of the theorem. $\qquad\square$

## A.5 PROOF OF THEOREM 3

First, we state two useful lemmas needed for the proof. We do not claim originality on those statements as those type of derivations may be found in literature on extremal events such as Embrechts et al. [2013]. As we could, however, not find the precise statements needed, we prove them now.

**Lemma 5.** *Let $q \in \mathbb{N}$ and for $e \in [E]$ let $C_1^e \overset{i.i.d}{\sim} \chi^2(k)$ and for $e \in [qE]$ let $C_2^e \overset{i.i.d}{\sim} \chi^2(k)$. For the random variables $Q_E = \max\limits_{e \in [E]} C_1^e$ and $W_E = \max\limits_{e \in [qE]} C_2^e$ it then holds that*

$$\lim_{E \to \infty} \mathbf{P}\left(\frac{W_E}{Q_E} = 1\right) = 1.$$

*Proof.* We prove the lemma by showing that for any $c > 1$ we have $\lim\limits_{E \to \infty} \mathbf{P}(W_E/Q_E > c) = 0$ and for any $c < 1$ that $\lim\limits_{E \to \infty} \mathbf{P}(W_E/Q_E < c) = 0$. The statement of the lemma then follows from a union bound over the events $\{W_E/Q_E \notin [1 - \frac{1}{m}, 1 + \frac{1}{m}]\}_{m \in \mathbb{N}}$. Hashorva et al. [2012] show that there exists a series $b_E$ such that for $E \to \infty$ we have $b_E \to \infty$ and $(W_E - b_E)$ converges to a distribution with support on $\mathbb{R}$. With that in hand we start by showing the case for $c > 1$. First, choose $\delta > 0$ such that $c > \frac{1+\delta}{1-\delta}$. With this we have that

$$\mathbf{P}\left(\frac{W_E}{Q_E} > c\right) \leq \mathbf{P}\left(\frac{W_E}{Q_E} > \frac{b_E(1+\delta)}{b_E(1-\delta)}\right)$$
$$\leq \mathbf{P}(W_E > b_E(1+\delta)) + \mathbf{P}(Q_E < b_E(1-\delta)).$$

We observe that for $E \to \infty$ the probability $\mathbf{P}(W_E > b_E(1+\delta)) = \mathbf{P}((W_E - b_E) > b_E\delta)$ converges to $0$ since $(W - b_E)$ converges to a distribution with support on $\mathbb{R}$ and $b_E \to \infty$. Analogue to this one may show that $\lim\limits_{E \to \infty} \mathbf{P}(Q_E < b_E(1-\delta)) = 0$. The case for $c < 1$ works analogue to $c > 1$. $\qquad\square$

**Lemma 6.** *For $e \in [E]$ let $W^e \overset{i.i.d}{\sim} \chi^2(k)$. Then*

$$\lim_{E \to \infty} \mathbf{P}\left(E^{\frac{2}{k}} \min_{e \in [E]} W^e > w\right) = e^{-w^{\frac{k}{2}} c_0},$$

*where $c_0 \in \mathbb{R}$ is a term constant in $w$. This also implies that the density function $f_E(w)$ of the random variable $\lim\limits_{E \to \infty} E^{\frac{2}{k}} \min\limits_{e \in [E]} W^e$ is given by*

$$f(w) = \frac{k}{2} w^{\frac{k}{2}-1} c_0 e^{-w^{\frac{k}{2}} c_0}.$$

*Proof.* The cumulative distribution function $F(w)$ of any $W^e$ is given by $F(w) = \tilde{c}_0 \gamma \left( \frac{k}{2}, \frac{w}{2} \right)$, where $\tilde{c}_0 := \frac{1}{\Gamma(\frac{k}{2})}$ is a term constant in $w$, Here $\Gamma$ is the gamma function and $\gamma$ is the lower incomplete gamma function defined for $s > 0, x > 0$ as

$$\gamma(s, x) = \int_0^x t^{s-1} e^{-t} dt.$$

Using the power series definition of the exponential term we can derive that

$$\gamma \left( \frac{k}{2}, \frac{w}{2} \right) = \left( \frac{w}{2} \right)^{\frac{k}{2}} \sum_{m \geq 0} \left( \frac{-w}{2} \right)^m \frac{1}{m!(\frac{k}{2} + m)} = \frac{2}{k} \left( \frac{w}{2} \right)^{\frac{k}{2}} + O\left( w^{\frac{k}{2}+1} \right).$$

Next, using the relation $F(w) = \tilde{c}_0 \gamma \left( \frac{k}{2}, \frac{w}{2} \right)$ and the equation above we conclude that for $c_0 := \frac{2}{k} \tilde{c}_0$ it holds that

$$\lim_{E \to \infty} \mathbf{P} \left( E^{\frac{2}{k}} \min_{e \in [E]} W^e > w \right) = \lim_{E \to \infty} \left( 1 - F\left( \frac{w}{E^{\frac{2}{k}}} \right) \right)^E$$
$$= \lim_{E \to \infty} \left( 1 - \frac{1}{E} c_0 w^{\frac{k}{2}} - O\left( E^{-\left(1 + \frac{2}{k}\right)} \right) \right)^E = e^{-c_0 w^{\frac{k}{2}}}.$$

The last equality uses the limit definition of the exponential function.

$\square$

**Proof of Theorem 3**

*Proof.* To identify when our method accepts $S$ as a set of potential causal parents, we have to understand the probability $\mathbf{P}(T_S > \min_{e \in [E]} Z_S^e / \max_{e \in [E]} Z_S^e)$, which is the key quantity for the hypothesis test $\phi_S$, defined in Equation (6), that our Algorithm 1 uses. For notational convenience we set $Z_{\min} := \min_{e \in [E]} Z^e$ and $Z_{\max} := \max_{e \in [E]} Z^e$. For this proof we assume without loss of generality that $\rho^v > \rho^w$. We are investigating the behavior of $\mathbf{P}(T(\boldsymbol{X}_E, \boldsymbol{Y}_E) > Z_{\min}/Z_{\max})$ for $E \to \infty$, the statements of the theorem will then follow by Markov's Inequality. By Lemma 4 we know that $P^v := \frac{1}{\rho^v} \|r_S^v\|_2 \sim \chi^2(n - |S|)$ and $Q^w := \frac{1}{\rho^w} \|r_S^w\|_2 \sim \chi^2(n - |S|)$ for $\rho^v := \sum_{u \in U} (\beta_u^1)^2 (\sigma_u^v)^2 + (\sigma_Y)^2$ and $\rho^w := \sum_{u \in U} (\beta_u^2)^2 (\sigma_u^w)^2 + (\sigma_Y)^2$. This allows us to rewrite $T(\boldsymbol{X}_E, \boldsymbol{Y}_E)$ as:

$$T(\boldsymbol{X}_E, \boldsymbol{Y}_E) = \frac{\min_{v \in [E_1], w \in [E_2]} (\min(\rho^v P^v, \rho^w Q^w))}{\max_{v \in [E_1], w \in [E_2]} (\max(\rho^v P^v, \rho^w Q^w))}.$$

To proof the first statement of the theorem we note that

$$\mathbf{P} \left( T(\boldsymbol{X}_E, \boldsymbol{Y}_E) > \frac{Z_{\min}}{Z_{\max}} \right) \leq \mathbf{P} \left( \frac{\min_{w \in [E_2]} \rho^w Q^w}{\max_{v \in [E_1]} \rho^v P^v} > \frac{Z_{\min}}{Z_{\max}} \right)$$
$$= \mathbf{P} \left( \frac{\rho^w}{\rho^v} \frac{\min_{w \in [E_2]} Q^w}{Z_{\min}} > \frac{\max_{v \in [E_1]} P^v}{Z_{\max}} \right).$$

By Lemma 5 it holds that

$$\lim_{E \to \infty} \mathbf{P} \left( \frac{\rho^w}{\rho^v} \frac{\min_{w \in [E_2]} Q^w}{Z_{\min}} > \frac{\max_{v \in [E_1]} P^v}{Z_{\max}} \right) = \lim_{E \to \infty} \mathbf{P} \left( \frac{\rho^w}{\rho^v} \frac{\min_{w \in [E_2]} Q^w}{Z_{\min}} > 1 \right). \tag{24}$$

For brevity we write $c := \frac{\rho^v}{\rho^w} = \frac{1}{I_S}$ and $Q := \min_{w \in [E_2]} Q^w$. Defining $f_E(x)$ as the density function of $E^{\frac{2}{k}} Z_{\min}$ and using

the result of Lemma 6 we continue with:

$$\lim_{E\to\infty} \mathbf{P}\left(\frac{Q}{Z_{\min}} > c\right) = \lim_{E\to\infty} \mathbf{P}\left(\frac{E^{\frac{2}{k}}Q}{E^{\frac{2}{k}}Z_{\min}} > c\right)$$

$$= \lim_{E\to\infty} \int_0^\infty f_E(x)\mathbf{P}\left(|[E_2]|^{\frac{2}{k}}Q > 2^{-\frac{2}{k}}cx\right)dx \tag{25}$$

$$= \int_0^\infty \frac{k}{2}x^{\frac{k}{2}-1}c_0 e^{-x^{\frac{k}{2}}c_0}e^{-\frac{1}{2}(cx)^{\frac{k}{2}}c_0}dx \tag{26}$$

$$= \int_0^\infty \frac{k}{2}x^{\frac{k}{2}-1}c_0 e^{-x^{\frac{k}{2}}c_0\left(1+\frac{1}{2}c^{\frac{k}{2}}\right)}dx$$

$$= -\frac{2}{c^{\frac{k}{2}}+2}e^{-x^{\frac{k}{2}}c_0\left(\frac{1}{2}c^{\frac{k}{2}}+1\right)}\Big|_0^\infty = \frac{2}{c^{\frac{k}{2}}+2} = \frac{2(I_S)^{\frac{k}{2}}}{2(I_S)^{\frac{k}{2}}+1}.$$

Here we use from (25) to (26) Lebesgue's dominated convergence theorem, which allows us to move the limit into the integral, and the limiting results from Lemma 6. To summarize, we have shown that

$$\mathbf{P}\left(T(\boldsymbol{X}_E,\boldsymbol{Y}_E) > \frac{Z_{\min}}{Z_{\max}}\right) \leq \frac{2(I_S)^{\frac{k}{2}}}{2(I_S)^{\frac{k}{2}}+1}.$$

The first statement of the theorem then follows by using this bound, together with the Markov's Inequality applied to

$$\mathbf{P}(\phi_S(\boldsymbol{X}_E,\boldsymbol{Y}_E)=0) = \mathbf{P}\left(\mathbf{P}\left(T(\boldsymbol{X}_E,\boldsymbol{Y}_E) > \frac{Z_{\min}}{Z_{\max}}\,\Big|\,\boldsymbol{X}_E,\boldsymbol{Y}_E\right) \geq \alpha\right).$$

To proof the second statement of the theorem we first note that

$$\mathbf{P}\left(T(\boldsymbol{X}_E,\boldsymbol{Y}_E) > \frac{Z_{\min}}{Z_{\max}}\right)$$
$$\geq \mathbf{P}\left(\frac{\min_{v\in[E_1],w\in[E_2]}(\rho^w \min(P^v,Q^w))}{\max_{v\in[E_1],w\in[E_2]}(\rho^v \max(P^v,Q^w))} > \frac{Z_{\min}}{Z_{\max}}\right),$$

making use of the assumption that $\rho^v > \rho^w$. The normality and independence Assumption 3 together with the additional mutual independence assumption of the collection $\{P^v,Q^w\}_{v\in[E_1],w\in[E_2]}$ allows us again to apply Lemma 5 and similar derivations to the ones following Equation (24) then lead to the conclusion that

$$\lim_{E\to\infty}\mathbf{P}\left(\frac{\min_{v\in[E_1],w\in[E_2]}(\rho^w \min(P^v,Q^w))}{\max_{v\in[E_1],w\in[E_2]}(\rho^v \max(P^v,Q^w))} > \frac{Z_{\min}}{Z_{\max}}\right) = \frac{(I_S)^{\frac{k}{2}}}{(I_S)^{\frac{k}{2}}+1}.$$

In summary this means that

$$\mathbf{P}\left(T(\boldsymbol{X}_E,\boldsymbol{Y}_E) > \frac{Z_{\min}}{Z_{\max}}\right) \geq \frac{(I_S)^{\frac{k}{2}}}{(I_S)^{\frac{k}{2}}+1}.$$

The second statement of the theorem follows if we combine the bound above together with a transformation of the bound given by the following Markov's Inequality:

$$1 - \mathbf{P}\left(\mathbf{P}\left(T(\boldsymbol{X}_E,\boldsymbol{Y}_E) > \frac{Z_{\min}}{Z_{\max}}\,\Big|\,\boldsymbol{X}_E,\boldsymbol{Y}_E\right) \geq \alpha\right)$$
$$= \mathbf{P}\left(1 - \mathbf{P}\left(T(\boldsymbol{X}_E,\boldsymbol{Y}_E) > \frac{Z_{\min}}{Z_{\max}}\,\Big|\,\boldsymbol{X}_E,\boldsymbol{Y}_E\right) \geq 1-\alpha\right)$$
$$\leq \frac{1}{1-\alpha}\left(1 - \mathbf{E}\left[\mathbf{P}\left(T(\boldsymbol{X}_E,\boldsymbol{Y}_E) > \frac{Z_{\min}}{Z_{\max}}\,\Big|\,\boldsymbol{X}_E,\boldsymbol{Y}_E\right)\right]\right).$$

$\square$

# B DATA GENERATION AND ADDITIONAL RESULTS

In this section we describe the precise data generation mechanisms used in our experiments from Section 6. Unless otherwise stated we fixed $E = 30$, $D = 6$ and $|S^*| = 2$. The data is generated from a linear structural equation model with different noise distributions, given by Equations (27)-(33) further below. Here $\mathcal{D}(\sigma)$ is a distribution with standard deviation $\sigma$ and zero mean. The specific choices of $\sigma$ for the individual experiments are specified in the following subsections. The graphical representation of the SEM is shown in Figure 7.

$$X_1 = \mathcal{D}(\sigma_1) \tag{27}$$
$$X_2 = X_1 + \mathcal{D}(\sigma_2) \tag{28}$$
$$X_3 = 0.3X_1 + \mathcal{D}(\sigma_3) \tag{29}$$
$$X_4 = 0.2X_3 + \mathcal{D}(\sigma_3) \tag{30}$$
$$Y = \beta_2 X_2 + \beta_3 X_3 + \mathcal{D}(\sigma_Y) \tag{31}$$
$$X_5 = 0.1X_2 + 0.3Y + \mathcal{D}(\sigma_5) \tag{32}$$
$$X_6 = 0.5Y + \mathcal{D}(\sigma_6) \tag{33}$$

**Effects of Non-Normal Noise.** The data for a single run within the noise misspecification experiment is generated in the following way. For each environment $e \in [E]$ we sample a vector of standard deviations $(\sigma_1^e, \cdots, \sigma_6^e)$, such that each entry is independently sampled from the uniform distribution on $[1, 5]$. The support entries of $\beta^e$, which are given by $S^* = \{2, 3\}$, are sampled in the same way. The data is then created with the additional relations defined through Equation (27)-(33). The noise distributions $\mathcal{D}(\sigma_1), \cdots, \mathcal{D}(\sigma_6)$ for the covariates are Gaussian with standard deviations as described above, while the target noise distribution $\mathcal{D}(\sigma_Y)$ is either a uniform, Student-t or Gaussian distribution (as indicated in the figures) with $\sigma_Y = 1.1$.

**Comparison with ICP** As L-ICP and ICP are similar in many regards, we chose for three simple experiments that highlights the differences. We set $\mathcal{N}(m, s)$ to be the standard normal distribution with mean $m$ and standard deviation $s$. In all three experiments we independently sample in $E = 100$ environments $n = 7$ observations $X_{1,i}^e \sim \mathcal{N}(0, \sigma)$, $X_{2,i}^e \sim \mathcal{N}(0, \sigma)$ and $Y_i^e = \beta^e X_{1,i}^e + \varepsilon_i^e$ with $\varepsilon_i^e \sim \mathcal{N}(0, 1)$.

In the *dense* setting we set $\beta^e = 1$ and sample in each environment $\sigma$ from a uniform distribution on $[1, 5]$.

In the *sparse* setting we set $\beta^e = 1$, as well as $s = 1$ for 99 out of the 100 environments. In the last environment we set $s = 3$.

In the ICP *violation* setting we follow the dense setting, but additionally sample $\beta^e$ independently from a uniform distribution on $[1, 5]$.

**Comparison with LiNGAM.** The data for a single run within the comparison to LiNGAM experiment is generated in the following way. For each environment $e \in [E] = [30]$ with $e \leq 15$ we set $(\sigma_1^e, \cdots, \sigma_6^e) = (2, \cdots, 2)$ and the support entries of $\beta^e$, which are given by $S^* = \{2, 3\}$, are set to $(1, 1)$. For $e > 15$ we set $(\sigma_1^e, \cdots, \sigma_6^e) = (c, \cdots, c)$ and $\beta_2^e = \beta_3^e = c$ for $c \in \{1, 1.2, 1.4, 1.5, 1.7, 1.8\}$ in the uniform noise case, $c \in \{1, 1.2, 1.4, 1.5, 1.6\}$ for the Gaussian noise case and $c \in \{1, 2, 5, 8\}$ for the scaled Student-t noise. To obtain the heterogeneity parameter $h$ we first apply Equation 8 to obtain $I_{\{2\}} = I_{\{3\}}$ (the relevant quantities to avoid false negatives) and then set $h = (I_{\{2\}})^{\frac{1}{4}} e^{1-(I_{\{2\}})^{\frac{1}{4}}}$ as also explained in Section 5.1. In this experiment we chose to not randomly sample the heterogeneity for a better control over it. The data is then created with the additional relations defined through Equation (27)-(33). The noise distributions $\mathcal{D}(\sigma_1), \cdots, \mathcal{D}(\sigma_6)$ and also $\mathcal{D}(\sigma_Y)$ are *all* uniform distributions for the uniform noise experiment, Gaussian distributions for the Gaussian noise experiment and scaled Student-t distributions for the last experiment. In all cases we set $\sigma_Y = 1$. Note that the experiment with the scaled Student-t distribution is found in Section B.1.

## B.1 ADDITIONAL EXPERIMENTS

**Adjusting the calibration under noise misspecification.** In Figures 1 and 2 of Section 6 we observed that under Student-t distributed noise, both the false negative and the false positive rate is adversely affected, in particular for larger sample sizes. In the following we show that this is not an inherent problem of our test statistic, but just due to the wrong calibration that assumes normal noise. If we adjust the target calibration $\alpha$ we can indeed recover a good performance as shown in Figure 6.

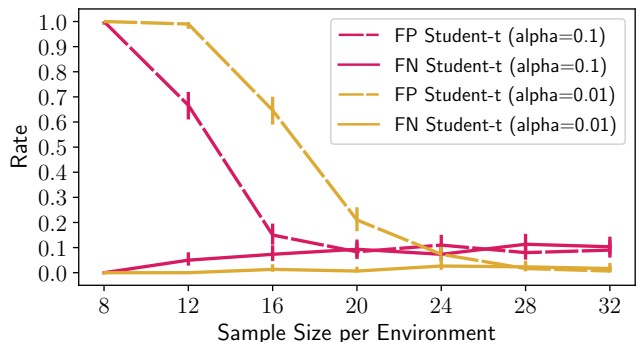

Figure 6: Under Student-t distributed noise, L-ICP achieves not the target calibration and this affects, in particular for larger samples, the performance. A near-optimal performance can be recovered if we adjust $\alpha$.

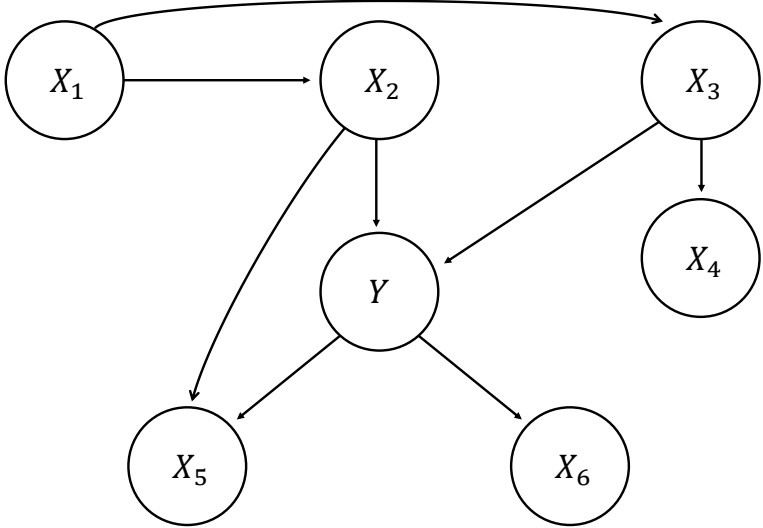

Figure 7: The structure of the linear structural equation model (SEM) we use in some experiments, ignoring the noise variables. The corresponding structural equations are given in (27)-(33).

How such an adjustment may be done in practice is unclear, and for that reason an important extension of L-ICP will be to find ways to calibrate the method without the normality assumption.

**Comparison with LiNGAM under Student-t noise.** To further study the effect of noise misspecification on L-ICPs performance we perform an additional comparison with LiNGAM when the noise comes from a scaled Student-t distribution. The results are shown in Figures 8a and 8b. We notice that if we set the degree of freedom to 3, so under a strong noise misspecification, L-ICP cannot recover good performance, also with strong heterogeneity of the environments. For 10 degrees of freedom, resulting in a weaker noise misspecification, L-ICP is again able to recover a good performance when there is sufficient heterogeneity in the environments.

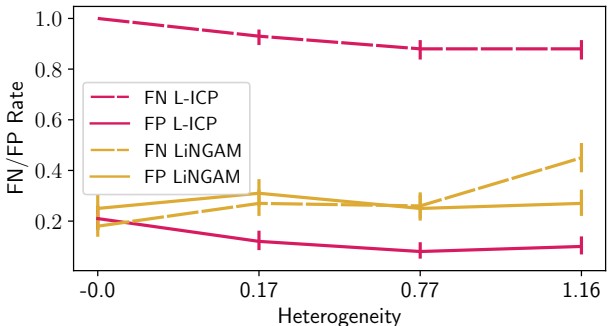 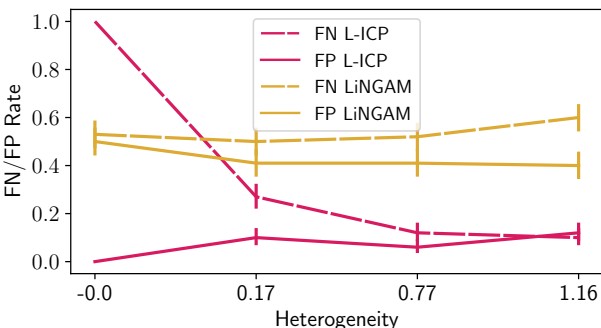

(a) The results when the scaled Student-t distribution has 3 degrees of freedom. While LiNGAM does make use of the strong non-normality, L-ICP can under the strong misspecifcation not recover a good performance, even when a lot of heterogeneity is present.

(b) The results when the scaled Student-t distribution has 10 degrees of freedom. As the noise misspecification is less strong, L-ICP can recover again good performance with the presence of heterogeneity.

## B.2 FULL RESULTS OF THE NETWORK DETECTION

The dynamical system is formally defined by the following set of equations, where the superscript $t$ indicates a time index.

$$X_1^{t+1} = 0.9X_1^t + 0.1X_2^t + \varepsilon_1^t \tag{34}$$

$$X_2^{t+1} = 0.28X_1^t - 0.01X_1^tX_3^t + 0.99X_2^t + \varepsilon_2^t \tag{35}$$

$$X_3^{t+1} = 0.01X_1^t(X_2^t - X_4^t) + 0.9733X_3^t + \varepsilon_3^t \tag{36}$$

$$X_4^{t+1} = 0.01X_1^t(X_3^t - 2X_5^t) + 0.9366X_4^t + \varepsilon_4^t \tag{37}$$

$$X_5^{t+1} = 0.02X_1^tX_4^t + 0.96X_5^t + \varepsilon_5^t \tag{38}$$

$$X_6^{t+1} = X_6^t + \varepsilon_6^t \tag{39}$$

Here $X_1^t, \cdots, X_5^t$ defines the Lorenz system, while $X_6^t$ is the random walk. Furthermore $\varepsilon_i^t$ for $1 \leq i \leq 6$ are random noise variables sampled independently from each other and past values from a standard normal distribution.

Here we report the full counts of the experiments of Section 6.1, additionally also when we use $n = 20$ samples for L-ICP. More precisely, let $\tilde{S}_{r,j}$ be the set of causal parents that L-ICP with given sample size $n$ reported in run $r$ for target covariate $j$, then we define $M_{i,j}^n := \sum_{r=1}^{500} \mathbf{1}\left\{i \in \tilde{S}_{r,j}\right\}$. The results of the experiments from Section 6.1 are then given by the following two matrices

$$M^{20} = \begin{bmatrix} 494 & 43 & 54 & 109 & 82 & 4 \\ 23 & 489 & 108 & 90 & 81 & 8 \\ 7 & 130 & 442 & 120 & 53 & 3 \\ 6 & 13 & 371 & 362 & 405 & 8 \\ 9 & 19 & 44 & 361 & 405 & 6 \\ 3 & 17 & 24 & 42 & 41 & 490 \end{bmatrix}, M^{25} = \begin{bmatrix} 498 & 68 & 87 & 110 & 96 & 2 \\ 56 & 470 & 193 & 93 & 72 & 5 \\ 3 & 238 & 337 & 108 & 58 & 3 \\ 3 & 28 & 320 & 189 & 189 & 6 \\ 3 & 27 & 97 & 227 & 227 & 3 \\ 3 & 17 & 43 & 44 & 38 & 496 \end{bmatrix}.$$

As PCMCI does not naturally group the environments, we run PCMCI over 30 individual intervals of length $n = 25$ in each of the 500 runs. The complete counts for PCMCI are:

$$N^{25} = \begin{bmatrix} 10778 & 1906 & 2209 & 1996 & 2033 & 1868 \\ 2422 & 12439 & 2035 & 1835 & 1830 & 1870 \\ 1766 & 2095 & 12816 & 2239 & 1852 & 1809 \\ 1875 & 1816 & 2201 & 12874 & 3273 & 1838 \\ 1909 & 1848 & 2000 & 3030 & 12980 & 1960 \\ 1857 & 1832 & 1851 & 1898 & 1879 & 9968 \end{bmatrix}$$

Based on the ground truth graph we picked a threshold of 1994 and report the edge from $i$ to $j$ if $N_{i,j}^{25} > 1994$.

