# OpenReview forum: "Invariant Causal Prediction with Local Models"
_auai.org/UAI/2024/Conference — UAI 2024 poster_

### Official Review · Reviewer_QAF4 · 2024-03-12

**Q2-1 Originality-Novelty:** 3
**Q2-2 Correctness-Technical Quality:** 3
**Q2-5 Clarity Of Writing:** 4

**Q1 Summary And Contributions:**

The authors present an extension of Invariant Causal Prediction (ICP) which exploits heterogeneity to infer the parents of a target variable. Instead of assuming the same linear functional relation between covariates and the target variable, this work allows for the parameters of this linear relationship to change between environments.

Given the data generation with assumed local linearity, the authors establish theoretical results on the control of the false positive rate and the false negative rate of detecting causal parents of a target of interest. A practical algorithm based on the aforementioned theory is then introduced, together with finite sample bounds on the false negative rate, which require additional technical assumptions. Finally, the proposed method is validated with numerical experiments.

**Q2-3 Extent To Which Claims Are Supported By Evidence:**

4: Excellent: all claims are supported by very convincing evidence (in the form of comprehensive experimental evaluation, rigorous mathematical proofs, detailed (pseudo-)code, precise references, well-motivated and realistic assumptions) and the authors deliver what they promise.

**Q2-4 Reproducibility:**

4: Excellent: key resources (e.g. proofs, code, data) are available and key details (e.g. proof sketches, experimental setup) are comprehensively described for competent researchers to confidently and easily reproduce the main results.

**Q3 Main Strengths:**

- The control over the false negative rate and the related finite sample results are highly interesting. ICP controls for the false positive rate and is known to be overly conservative, which is directly addressed in this paper. The assumptions that are required for these results are worth being discussed and may be questioned, but I view the shown results as a welcome starting point to address this issue.
- The approach of the paper is well-motivated and its contribution therefore quite clear. The assumption of local linearity as opposed to global linearity is natural to move towards more general function classes, underlined by the example of identifying a nonlinear dynamical system. Overall this work is a solid extension of the ICP framework that addresses some of its limitations.
- The paper is very nicely structured and well written. The presentation helps to understand the motivation and concepts are introduced concisely and in an order that supports understanding.

**Q4 Main Weakness:**

- Some of the assumptions made throughout the paper deserve a more thorough discussion. Specifially Assumptions 2 and 4, which I will expand upon below in Q5.

**Q5 Detailed Comments To The Authors:**

- The main point I would like to be discussed in more detail is Assumption 4 pertaining the the independence of covariates. You essentially have to assume independence of all covariates for Theorems 2 and 3, correct? You mention that prior works do something similar, but this still feels slightly glossed over. I understand that often strong assumptions are required to establish finite sample results and given the experimental results where you apparently don't need these assumptions for the false negative rate to decline in practice (a strength!), I think discussing this more openly would benefit the work overall.
- Assumption 2 is required for your proof and you discuss it in some detail, but could you expand upon the point of polynomial dependence on $Y^e$? Which other types of regularity assumptions are required for your results to still hold? How strong do they have to be?
- The population results you present require and arbitrarily large amount of samples per environment. Do you also require an arbitrarily large amount of environments? Is it then implicit, that these environments are maximally heterogeneous?
- While the assumptions required for Theorems 2 and 3 are strong, characterizing these results in terms of a degree of heterogeneity is very interesting. Could you comment on what different values of $h_1$ and $h_2$ correspond to? Do we require an intervention on each variable or maybe multiple interventions on each variable with sufficient strength?
- One way the assumption of local linearity could also be viewed is by extending the class of permissible interventions compared to ICP. In the original paper, the authors only provide identifiability results for do-interventions and noise interventions, whereas this might be framed as allowing a subclass of interventions that must maintain the parents of a variable, as well as a linear functional relationship, but is otherwise free. Perhaps this introduces a novel perspective for comparison to other works.
- Did you perform an experimental comparison to ICP? Your method seems less conservative and I would expect you outperform ICP on some settings.
- I think the experiment with the nonlinear system is a great showcase of what your approach can deal with and is an excellent motivational use-case for your method. Both handling a nonlinear system, as well as identifying a whole graph as opposed to the parents of a single node are very intriguing.

Other comments:

-  I am not an expert on statistical testing, so I cannot comment in depth on the proposed procedure from this perspective. From my basic understanding, I cannot identify major issues, but the other reviewers will have to comment on this.

**Q9 Complying With Reviewing Instructions:**

Yes

---

> ### Author Rebuttal · Authors · 2024-04-08
>
> We would like to thank the referee for the high quality feedback, and address the specific remarks point-by-point:
>
> **The main point I would like to be discussed in more detail is Assumption 4... You essentially have to assume independence of all covariates for Theorems 2 and 3, correct? ... I think discussing this more openly would benefit the work overall.**
>
> For those theorems we indeed assume independence of all covariates, and additionaly that the target does not have any children. This are indeed very strong assumptions. Nevertheless, these provide a natural starting point to understand the finite sample properties of the proposed method. Furthermore, we experiment with violations to this in various ways to get further insights. Given the extra space in the final paper we are happy to move the full set of assumptions in the main paper. As we then have access to the formal definitions it will also be easier to contextualize them with respect to related work.
>
> **Assumption 2 is required for your proof and you discuss it in some detail, but could you expand upon the point of polynomial dependence on $Y^e$? Which other types of regularity assumptions are required for your results to still hold? How strong do they have to be?**
>
>  Given the extra space we are happy to further elaborate on the polynomial dependence assumption, which will also serve towards a sketch proof: Under the setting of the theorem it is sufficient to show that the variance of residuals in one environment $V_1$ is not the same as the variance of residuals in another environment $V_2$. We then analyse the equation $V_1=V_2$ in terms of the parameters $\beta^e$. Fixing $V_2$ and showing that $V_1$ is a polynomial equation in $\beta^e$ (that comes from the polynomial dependence) we conclude that  $V_1=V_2$ can only be fulfilled for at most finitely many $\beta^e$. Regarding the above argument, we require a function class such that $V_1=constant$ has still at most countably many solutions with respect to $\beta^e$. Any sufficiently smooth function class could possibly lead to such a result.
>
> **The population results you present require and arbitrarily large amount of samples per environment. Do you also require an arbitrarily large amount of environments? Is it then implicit, that these environments are maximally heterogeneous?**
>
> In all theorems we require only a minimum of two heterogeneous environments. The infinite environment case leads actually to similar results as shown in Theorem 3. Furthermore we require minimal heterogeneity in the sense that environments may be distributionally arbitrarily close to each other, but are not allowed to be the same.
>
> **While the assumptions required for Theorems 2 and 3 are strong (...) Could you comment on what different values [of heterogeneity] correspond to? Do we require an intervention on each variable?:**
>
> Let $X^e$ be a parent of $Y^e$, with $e \in \{v,w\}$, and $\sigma^e$ be the standard deviation of $X^e$ and $\beta^e$ the structural parameter belonging to $X^e$. Under those assumptions, and in a no-noise setting, the heterogeneity belonging to $X^e$ is $\frac{(\beta^v \sigma^v)^2 }{(\beta^w \sigma^w)^2 }$ (or the inverse, whichever is smaller). The total heterogeneity is then the minimal heterogeneity over all covariates. Generally we indeed require that every covariate has at least one intervention, but those interventions may be performed all at once, so that we only require at least two environments.
>
> **One way the assumption of local linearity could also be viewed is by extending the class of permissible interventions compared to ICP:**
>
>  This is a very nice point. In fact, we had such a remark in an earlier draft, but was unfortunately removed due to space limitations: locality not only makes the model more flexible but also extends the notion of heterogeneity in environments.
>
> **Did you perform an experimental comparison to ICP? Your method seems less conservative and I would expect you outperform ICP on some settings:**
>
> This is a good intuition that we did not anticipate. We explored this in a new set of experiments, that we will incorporate in the final version of the paper, in three different settings: two settings where the assumptions of ICP are met and have either dense or sparse heterogeneity in the environments, and one setting where the assumptions of ICP are violated. The results can be seen in https://ibb.co/WFKBkhP. Note that the Bonferroni correction used in the tests in ICP Method II renders it very conservative (very low FPR) in comparison with LoLiCaP, showing that the max/min test we proposed may be useful beyond our setting. Furthermore, LoLiCaP's FNR is lower than that of ICP for all but the sparse setting, so, it can outperform ICP even when its assumptions are met.
>
> **Remark:** Based on the other reviewers feedback we also plan to incorporate comparisons with further baselines, as outlined in our official comment.

---

### Official Review · Reviewer_tAmY · 2024-03-13

**Q2-1 Originality-Novelty:** 3
**Q2-2 Correctness-Technical Quality:** 3
**Q2-5 Clarity Of Writing:** 3

**Q10 Ethical Concerns:**

No.

**Q1 Summary And Contributions:**

The authors introduce a version of invariant causal prediction on heterogeneous linear environments with varying effect coefficients. They discuss theoretical results as well as a method for parent identification in their setting.

**Q2-3 Extent To Which Claims Are Supported By Evidence:**

3: Good: the main claims are supported by convincing evidence (in the form of adequate experimental evaluation, proofs, (pseudo-)code, references, assumptions).

**Q2-4 Reproducibility:**

4: Excellent: key resources (e.g. proofs, code, data) are available and key details (e.g. proof sketches, experimental setup) are comprehensively described for competent researchers to confidently and easily reproduce the main results.

**Q3 Main Strengths:**

Both the theory and the experiment section are well-written and thorough, and contain interesting ideas.

**Q4 Main Weakness:**

Theorems 2 and 3 rely on very strong assumptions that are only stated in the appendix. To be fair, the authors mention this in the main text but it would be more transparent to state these assumptions, at least colloquially. The paper could also do with another example of a setting where locally linear models might be reasonably used.

**Q5 Detailed Comments To The Authors:**

•	Finding causal parents of a disease: While I agree that there is a lot of heterogeneity in this example, I am not so sure it is the type of heterogeneity that fits to your assumptions. I would either imagine heterogeneous effect due to confounding factors or due to the presence of unobserved factors that change the noise distributions between countries in this example. Can you give a plausible example in which the mechanisms by which the parents cause the disease differ from country to country?

•	Finding causes of a mechanism shift: I like this example more, even though here one should perhaps add that samples should be sufficiently far apart in time to avoid autocorrelated noise which would break your assumptions.

•	I think that the theory section is written in a very clear and understandable way despite the technicality of some the assumptions, e.g. Assumption 2. The proofs in the supplement are quite extensive so that I have not been able to check them in full detail. Due to their technicality, they would deserve a more extensive review than this reviewing format allows for. This is unfortunate but hardly the authors’ fault.

•	Theorem 2: typo: assumption -> assumptions

•	Theorem 2: It would be good to at least give a hint as to what the additional assumptions are. Looking at Assumption 4 in the appendix, they look very strong. You mention that they are strong in the main text but it would help to state them at least colloquially.

•	The dynamic network experiment is an interesting illustration, even though the task to decide on a good split of intervals would seem extremely hard if this was a real-world example in which a ground-truth is not known. I understand that this might be beyond the scope of this work, but a comparison of this to similar methods on real data would be very desirable in future work.

•	Changing distribution of the target noise: I find this explanation confusing, can you elaborate what you mean by independence of the environment index $e$ and $\theta^e$ which seems to depend on $e$?

**Q9 Complying With Reviewing Instructions:**

Yes

---

> ### Author Rebuttal · Authors · 2024-04-08
>
> We would like to thank the referee for the high quality feedback, and address the specific remarks point-by-point:
>
> **Theorems 2 and 3 rely on very strong assumptions that are only stated in the appendix (...) The paper could also do with another example:**
>
> Given the extra space of the final submission we are happy to move the assumptions from the appendix to the main paper to better contextualize them. In addition, we plan to include the following extra example (with additional details): Our task is to identify important factors that drive fluctuations in stock prices and market volatility. For that task we identify important legislation regarding regulations of the stock market. One may assume that both, the distributions of the important factors, as well as the mechanism between factors and stock prices and market volatility can change after an important legislation was put into place. We chose our environments now as the time between those legislations.
>
> **Finding causal parents of a disease:...**
>
> This is a valid point. Generally we consider the health example rather as an example of a changing covariate distributions, than of a changing mechanism example. However, important unobserved factors may (approximately) also be absorbed as a change of mechanism: consider the observed phenomenon that the mortality rate under covid was much lower in countries with a good healthcare system. If we would not observe the quality of the healthcare system, this factor may be absorbed in the change of the mechanism between covid and mortality in different countries: it may then translate to a change of importance of how covid affects mortality across different countries.
>
>  **Finding causes of a mechanism shift (...) samples should be sufficiently far apart in time (...):**
>
>  Indeed, working with time-series adds another layer of complication to the methodology, which we did not touch upon, but we are happy to include your suggested remark.
>
>  **The dynamic network experiment is an interesting illustration, even though the task to decide on a good split of intervals would seem extremely hard if this was a real-world example:**
>
> The decision on the split is certainly crucial, but in our experience less problematic than one might anticipate: note that, as also mentioned in the paper, the methodology will return the empty set if the interval is too large or too small. So all we have to identify is an interval size in which LoLiCaP does (generally) not return the empty set. We actually applied this exact idea on real world data from lithography processes (which gave us the motivation for the paper). We obtained "seemingly" meaningful results, but these are not included due to lack of actual ground-truth and confidentially of the data.
>
> **Changing distribution of the target noise... can you elaborate what you mean by independence of the environment index $e$ and $\theta^e$  which seems to depend on $e$?**
>
> This is indeed confusing, we will remove the index $e$ from $\theta^e$. It was just supposed to indicate that this is the specific $\theta$ we observed together with environment $e$, i.e. we observe joint entities $(e,\theta)$.
>
> **Remark:** Based on the other reviewers feedback we also plan to incorporate comparisons with further baselines, as outlined in our official comment.

---

### Official Review · Reviewer_Efh2 · 2024-03-15

**Q2-1 Originality-Novelty:** 2
**Q2-2 Correctness-Technical Quality:** 2
**Q2-5 Clarity Of Writing:** 3

**Q1 Summary And Contributions:**

The authors present an extension of invariant causal prediction (ICP) in which
the causal mechanism is assumed to be linear but allowed to depend on the
environment E and the goal is to identify the set of causal parents S^* among D
predictors of a response Y. Under the assumption that the error does not depend
on E but rather follows a fixed distribution F^*, the authors propose to test
whether the residual variance varies with E. Critically, their test hinges on a
normality assumption of the error whose impact is studied empirically. Finally,
the authors compare the performance of their proposed methods with LiNGAM and
showcase their method in a dynamical systems simulation.

**Q2-3 Extent To Which Claims Are Supported By Evidence:**

2: Fair: the main claims are somewhat supported by evidence (but the experimental evaluation may be weak, or does not match entirely with the claims, important baselines may be missing, proofs contain important ideas but lack rigor, algorithmic details are only discussed superficially, references are imprecise, assumptions are not sufficiently motivated or explicated, etc.).

**Q2-4 Reproducibility:**

3: Good: key resources (e.g. proofs, code, data) are available and key details (e.g. proofs, experimental setup) are sufficiently well-described for competent researchers to confidently reproduce the main results.

**Q3 Main Strengths:**

* The paper is written well and accessibly and the extension is motivated
  clearly.

* Assumptions and theoretical results are clearly stated.

* The authors provide code and notebooks for reproducibility.

**Q4 Main Weakness:**

I provide more detail under Q5.

* The level violations for heavy-tailed noise are rather worrisome.

* Comparisons with LiNGAM are of debatable usefulness and no other baselines are
  considered.

**Q5 Detailed Comments To The Authors:**

* At first, the name "locally linear" seems slightly misleading: Given E = e,
  the paper assumes linear models. But the name method does not use locally
  linear methods, such as smoothers. The authors could consider motivating their
  choice of name for the proposed method a bit more.

* What happens in the presence of hidden variables (i.e., violations of
  Assumption 1 eq. (3))? The original approach proposed in Peters et al (2016),
  under the assumption that there exists an invariant set among the observed
  ancestors, has a coverage guarantee for returning ancestors.

* Test proposed test assumes normality -- is there any way to generalize this?
  This warrants some discussion in my opinion, especially in connection to the
  level violations under heavy-tailed noise distributions. Have the authors
  considered asymmetric noise distributions too? I do appreciate the assumption
  is needed to obtain finite sample results, but the lack of robustness casts
  some doubt on how useful this method can be when applied to real data.

* Is there a way to compare against a version of the original ICP algorithm? I
  suspect ICP will simply return the empty set, since the causal mechanism
  changes with the environments, but this would be a good sanity check.

* LiNGAM assumes non-Gaussian noise, while the authors assume Gaussian noise.
  The authors were of course aware of this and decided using uniform noise in
  one of their experiments. However, the simulations have shown LOLICAP to be
  conservative in this setting (which can also be seen in Fig 3/4). It would be
  nice to see the results for the student t setting in the appendix. Is there no
  other method the authors could compare with (see also my above comment)?

* Calling Section 6.1 an "application" is very misleading because fully
  synthetic data is used. Perhaps, the authors could motivate their example as a
  "hypothetical application" or a "potential field of application"?

* Figure 6 and Appendix C.1 gives a rather hand-wavy "re-calibration" for alpha
  in the case of Student t noise without guarantees or proper arguments. The
  authors ought to consider removing this or discussing this more precisely.

* The method only works for discrete environment variables. Is there a way to
  generalize this to continuous environments?

Minor:

* "[...] finding a full causal graph is possible in our proposed setting if no
  covariate is directly affected by the environment index, but only indirectly
  through changing structural parameters." could be phrased more precisely.

* The introduction is rather short: Perhaps one could move some of the
  motivating examples from Section 3 there?

* Plotting false positive and negative rate in one plot is suboptimal due to the
  different scaling. It makes Figures 3, 4, 6 rather hard to read.

**Q9 Complying With Reviewing Instructions:**

Yes

---

> ### Author Rebuttal · Authors · 2024-04-08
>
> We would like to thank the referee for the high quality feedback, and address the specific remarks point-by-point:
>
> **Problematic Level violations for heavy-tailed noise:** The proposed over-arching methodology requires: i) estimation of parameters per environment; ii) testing for residual distribution differences across environments. Our specific choices for (i) and (ii) provide an objective way to study the potential of the overarching methodology. The least-squares choice for (i) (sensitive of outliers) might be replaced by robust regression methods (e.g., M-estimators with the appropriate influence functions, e.g, with a Huber-loss). Tests for heterogeneity (ii) can be calibrated by permutation as in [1] (e.g., by permuting the residuals over all environments and contrasting the test statistic in the permuted and unpermuted data. Alternatively, Levene's test for equal variances [2] may be used as in [3]. We will further discuss these interesting avenues in Section 7.
>
> **Comparison with additional baselines - in particular ICP:** This is certainly a valid point, and while the current experiments show we are not solving a trivial task, comparison to methods not explicitly making use of heterogeneity can be deemed unfair towards them.
> To further contextualize our contributions we will include and discuss the following experiments:
>
> i) Comparison between LoLiCaP and ICP (Method II from their paper) in three settings: two settings where the assumptions of ICP are met and have either dense or sparse heterogeneity in the environments, and one setting where the assumptions of ICP are violated. The results can be seen in https://ibb.co/WFKBkhP. Note that the Bonferroni correction used in the tests in ICP-II renders it very conservative (very low FPR) in comparison with LoLiCaP, showing that the max/min test we proposed may be useful beyond our setting. Furthermore, LoLiCaP's FNR is lower than that of ICP for all but the sparse setting, so, it can outperform ICP even when its assumptions are met.
>
> ii) We add a LiNGAM comparison with a scaled-student-t noise distribution, once for $t=3$ and once for $t=10$ degrees of freedom. While for $t=3$ also increased heterogeneity does not fix the problem due to model-mismatch, for $t=10$ LoLiCaP has good performance again, see also https://ibb.co/PYQMh4x.
>
>  iii) Based on the other reviewers feedback we also compare against a baseline on the Lorenz-data set, as outlined in our official comment.
>
> **The name "locally linear":** This terminology can be misinterpreted as the use of local polynomial regression, which was not our intention. On the other hand, if the environments are embedded in a metric space the collection of coefficients $\{\hat\beta^e\}_{e\in[E]}$ can be interpreted as fitted regressogram - which is a crude local regression method. To avoid confusion we are exploring different terminology options. As the overarching methodology is not specific to linear regression “Localized Invariant Causal Prediction” (LICP) might be suitable. We are grateful for other suggestions you may have.
>
> **The presence of hidden variables:** Although ICP can have guarantees in the presence of hidden variables in benign settings, it will return in that case ancestors rather than direct parents. LoLiCaP, on the other hand, will in general return the empty set in presence of most hidden variables: together with the mechanism change the hidden variable introduces a noise term in the target that changes between environments and cannot be explained by any subset of covariates. Nevertheless, under stringent assumptions (e.g., limited heterogeneity contribution of hidden variables) and modifications of the procedure it might be possible to cope with such scenarios.
>
> **Calling Section 6.1 an "application" is very misleading:**  We agree. The term application was not intended to refer to a real-world setting and associated data. We will name the section simply "Network Detection in Dynamical Systems"
>
> **Re-calibration for alpha:** The experiment leading to Figure 6 in appendix C.1 is not meant as a possible fix to the problem, but is merely supposed to show that both the false positive and negative rate are negatively affected by the wrong calibration. We are happy to make this context clearer.
>
> **Extension to continuous environments:** A similar extension was investigated in [3]. While our setting is still meaningful, our specific heterogeneity test has to be adapted, for example with general conditional independence tests, and possibly require extra smoothness assumptions.
>
> [1] Stoepker, I. V., Castro, R. M., Arias-Castro, E., and van den Heuvel, E. Anomaly Detection for a Large Number of Streams: A Permutation-Based Higher Criticism Approach. JASA, 2024
>
> [2] Brown, M. B., and Forsythe, A. B. “Robust Tests for the Equality of Variances.” JASA, 1974
>
> [3] Heinze-Deml, C., Peters, J. and Meinhausen, N. Invariant causal prediction for nonlinear models. J. Causal Inference, 2017

---

### Official Review · Reviewer_RKua · 2024-03-21

**Q2-1 Originality-Novelty:** 2
**Q2-2 Correctness-Technical Quality:** 3
**Q2-5 Clarity Of Writing:** 2

**Q10 Ethical Concerns:**

None.

**Q1 Summary And Contributions:**

The authors consider the problem of identifying the parents of a target variable $Y$ from a set of variables $X$ in the observational setting, where we may have access to multiple environments and heterogeneous data. The proposed method is mainly based on invariant causal prediction (ICP, Peters et al. 2016). However, compared with ICP, the proposed method allows the causal mechanism (i.e., coefficients) of $Y$ to be different across different environments. The authors provide theoretical guarantee of their proposed algorithm, and test the performance of their algorithm on synthetic data.

**Q2-3 Extent To Which Claims Are Supported By Evidence:**

2: Fair: the main claims are somewhat supported by evidence (but the experimental evaluation may be weak, or does not match entirely with the claims, important baselines may be missing, proofs contain important ideas but lack rigor, algorithmic details are only discussed superficially, references are imprecise, assumptions are not sufficiently motivated or explicated, etc.).

**Q2-4 Reproducibility:**

3: Good: key resources (e.g. proofs, code, data) are available and key details (e.g. proofs, experimental setup) are sufficiently well-described for competent researchers to confidently reproduce the main results.

**Q3 Main Strengths:**

1. The paper is well organized, and the problem is well motivated through examples.
2. The mathematical formulation of the problem is clear, and the notations are easy to follow.
3. The theoretical results and the model/identifiability assumptions are clearly stated.

**Q4 Main Weakness:**

1. The paper lacks (theoretical and practical) comparison with baseline methods, especially ICP. I would suggest the authors use one paragraph to summarize the differences between their proposed method and ICP (model assumptions, identifiability assumptions, algorithms, etc).
2. The simulation setting may not align with the claims in the theoretical part, see Question 2 below.

**Q5 Detailed Comments To The Authors:**

**Questions**:
1. The authors mentioned in Theorem 1 that "they mainly require that the noise distributions of all covariates are homogeneous, and heterogeneity is only introduced through changing structural parameters". However, this is not the case in the simulation, where the model is fixed, and the noise distributions are different across environments. If this is the case then this also satisfies the assumptions of ICP. Please correct me if I'm wrong.
2. Regarding Assumption 2, it seems that the authors assume that the causal mechanisms of all $X$ variables (i.e., $f_d$) are the same across environments, but not for $Y$ variables. Is there a particular reason/motivation for this?
3. Have the authors considered other methods in the simulation? Methods like ICP and JCI could also return a parent set. Besides, since the data is generated from an underlying SCM among all variables, is it possible to apply causal discovery algorithms on all ($X$ and $Y$) variables, and then return the parents of $Y$?
4. In the simulation there are $E=30$ environments. However, from my understanding the first 15 environments have the same underlying model. I am wondering if this is the case, and why they cannot be merged into a single environment.

**Q9 Complying With Reviewing Instructions:**

Yes

---

> ### Author Rebuttal · Authors · 2024-04-08
>
> We would like to thank the referee for the high quality feedback, and address the specific remarks point-by-point:
>
> **The paper lacks (theoretical and practical) comparison with baseline methods, especially ICP. I would suggest the authors use one paragraph to summarize the differences between their proposed method and ICP (model assumptions, identifiability assumptions, algorithms, etc:**
>
>  This is certainly a valid point. Note that comparison with methods that do not explicitly make use of the heterogeneity (when present) can be seen as unfair: By increasing the hetereogenity one can easily ensure LoLiCaP outperform those competitors. That being said we will add a comparison between LoLiCaP and ICP (Method II from their paper) in three settings: two settings where the assumptions of ICP are met and have either dense or sparse heterogeneity in the environments, and one setting where the assumptions of ICP are violated. The results can be seen in https://ibb.co/WFKBkhP. Note that the Bonferroni correction used in the tests in ICP Method II renders it very conservative (very low FPR) in comparison with LoLiCaP, showing that the max/min test we proposed may be useful beyond our setting. Furthermore, LoLiCaP's FNR is lower than that of ICP for all but the sparse setting, so, it can outperform ICP even when its assumptions are met.
>
> **The authors mentioned in Theorem 1 that "they mainly require that the noise distributions of all covariates are homogeneous, and heterogeneity is only introduced through changing structural parameters". However, this is not the case in the simulation, where the model is fixed, and the noise distributions are different across environments. If this is the case then this also satisfies the assumptions of ICP. Please correct me if I'm wrong.**
>
> The noise distribution is given by independent standard normal noises as specified after Equation (37), and thus the same across different intervals. The reviewer is right that the structural equation is also fixed, but note that it is a non-linear model. This non-linearity implies that the local linear approximations to this structural equation will change per interval, so from our local linearity view the parameters do change. This is also the reason why ICP is not expected to work, as also shown in the additional experiments.
>
> **Regarding Assumption 2, it seems that the authors assume that the causal mechanisms of all $X$ are the same across environments, but not for $Y$ variables. Is there a particular reason/motivation for this?**
>
> This is well spotted and just a notational mistake. We will add an environment index to those mechanisms.
>
> **Have the authors considered other methods in the simulation? Methods like ICP and JCI could also return a parent set. Besides, since the data is generated from an underlying SCM among all variables, is it possible to apply causal discovery algorithms on all $X$ and $Y$ variables, and then return the parents of $Y$**
>
> As explained above and illustrated above we do not expect that ICP returns a parent set when heterogeneity is present. That being said, we add a comparison against PCMCI [1] on the Lorenz-data set with $n=25$. This method finds a Markov set of nodes for all nodes with a PC-type algorithm and then removes false positive edges with a second conditional independence test on the lagged variables. For a fair comparison we use a partial correlation test for PCMCI on the same intervals as for LoLiCaP. Providing a summary graph from the edges PCMCI found with the thresholding we used for LoLiCaP ended in a trivial (fully connected) graph. For a meaningful comparison, and erring on the side of advantage for PCMCI, we chose a clairvoyant threshold for PCMCI. This resulted in the graph https://ibb.co/6gKqZK0. PCMCI has, as LoLiCaP, one false negative, but two less false positives. On the other hand, we are clairvoyantly choosing a threshold to optimize PCMCI performance and furthermore PCMCI relies explicitly on temporal information, while LoLiCaP does not.
>
> [1] Runge, J., Nowack, P., Kretschmer, M., Flaxman, S. and Sejdinovic, D. Detecting and quantifying causal associations in large nonlinear time series datasets, Science Advances, 2019
>
> **In the simulation there are $E=30$ environments. However, from my understanding the first 15 environments have the same underlying model. I am wondering if this is the case, and why they cannot be merged into a single environment.**
>
> You are correct. We made that choice to closer match the setting of Theorem 2. If one has a-priori knowledge that the environments are the same (distributionally), merging them should give better results: that would increase the sample size and we know that the power of our test converges exponentially fast to one in sample size.

---

### Official Review · Reviewer_wGaM · 2024-03-22

**Q2-1 Originality-Novelty:** 3
**Q2-2 Correctness-Technical Quality:** 3
**Q2-5 Clarity Of Writing:** 4

**Q1 Summary And Contributions:**

This paper studies a new version of Invariant Causal Prediction, where in each environment, there is a linear model, but the coefficients may be different in each environment. This heterogeneity can be leveraged to identify causal parents of the target variable, and the false positive/negative rates are analysed theoretically and experimentally. An example of a  setting where this method can be useful is when data from a dynamical system is segmented into short intervals.

**Q2-3 Extent To Which Claims Are Supported By Evidence:**

3: Good: the main claims are supported by convincing evidence (in the form of adequate experimental evaluation, proofs, (pseudo-)code, references, assumptions).

**Q2-4 Reproducibility:**

4: Excellent: key resources (e.g. proofs, code, data) are available and key details (e.g. proof sketches, experimental setup) are comprehensively described for competent researchers to confidently and easily reproduce the main results.

**Q3 Main Strengths:**

- The paper is well-written, building up the story in a logical order.
- Its contribution is an interesting advance over ICP, and its properties are discussed in an insightful way.
- The theoretical portion of the contribution is technically solid.
- The experiments are chosen well to illustrate several aspects of the algorithm's performance.

**Q4 Main Weakness:**

The experiments only compare the new method to LiNGAM. Experimental comparison to some of the other algorithms listed in the related work would be a good addition.

**Q5 Detailed Comments To The Authors:**

Questions:
- Below assumption 1, "It ensures the environment does not act as a confounder between covariates and target.": Can you elaborate how it ensures this? Assumption 1 does allow $\epsilon_i^e$ to depend on $\epsilon_j^{e'}$ in a different environment.

Minor comments:
- Introduction, "a locally linearity assumption": "local"
- The definition of causal parents $S^{*}$ (near equation (1)) is circular: it is defined here in terms of the nonzero coefficients in (1), but without the concept of causal parents, we can't say whether (1) is a structural equation or just a linear regression equation.
- "(convention that ...)": make this into a sentence
- below Lemma 1, "the above null hypothesis": which one?
- Assumption 2, "where $f_d$ is a polynomial of finite degree in $X_y^e$ ..., but otherwise arbitrary": I'm not quite sure what this means. Is it that if $y$ is a parent, then $f_d(\cdot) = \sum_{k=0}^K (X_y^e)^k g_d^{(k)}(X^e_{other})$, where the $g_d^{(k)}$ are arbitrary functions of the other parents?
- Section 7, under "The role of locality": "i" missing before "n some scenarios"

---

UPDATE: I'd like to thank the authors for their responses, which addressed my (mostly minor) concerns. I will keep my positive score.

**Q9 Complying With Reviewing Instructions:**

Yes

---

> ### Author Rebuttal · Authors · 2024-04-08
>
> We would like to thank the referee for the high quality feedback, and address the specific remarks point-by-point:
>
> **The experiments only compare the new method to LiNGAM. Experimental comparison to some of the other algorithms listed in the related work would be a good addition.**
>
>  This is certainly a valid point. While LiNGAMs performance shows that we are not solving a trivial task, comparing our approach to methods that do not explicitly make use of the heterogeneity (when present) can be seen as unfair: By increasing the hetereogenity one can easily ensure LoLiCaP outperforms those competitors. On the other hand we are not aware of other methods that use heterogeneity under arbitrary mechanism shift, one of the main motivations for this work. That being said we performed and will add two additional experiments to further contextualize the performance of LoLiCaP:
>
> (i) Comparison between LoLiCaP and ICP (Method II from their paper) in three settings: two settings where the assumptions of ICP are met and have either dense or sparse heterogeneity in the environments, and one setting where the assumptions of ICP are violated. The results can be seen in https://ibb.co/WFKBkhP. Note that the Bonferroni correction used in the tests in ICP Method II renders it very conservative (very low FPR) in comparison with LoLiCaP, showing that the max/min test we proposed may be useful beyond our setting. Furthermore, LoLiCaP's FNR is lower than that of ICP for all but the sparse setting, so, it can outperform ICP even when its assumptions are met.
>
> (ii) We perform PCMCI [1] on the Lorenz-data set with $n=25$. PCMCI finds a Markov set of nodes for all nodes with a PC-type algorithm and then removes false positive edges with a second conditional independence test on the lagged variables. For a fair comparison we use a partial correlation test for PCMCI on the same intervals as for LoLiCaP. Providing a summary graph from the edges PCMCI found with the thresholding we used for LoLiCaP ended in a trivial (fully connected) graph. For a meaningful comparison, and erring on the side of advantage for PCMCI, we chose a clairvoyant threshold for PCMCI. This resulted in the graph https://ibb.co/6gKqZK0. PCMCI has, as LoLiCaP, one false negative, but two less false positives. On the other hand, we are clairvoyantly choosing a threshold to optimize PCMCI performance and furthermore PCMCI relies explicitly on temporal information, while LoLiCaP does not.
>
> [1] Runge, J., Nowack, P., Kretschmer, M., Flaxman, S. and Sejdinovic, D. Detecting and quantifying causal associations in large nonlinear time series datasets, Science Advances, 2019
>
> **Below assumption 1, "It ensures the environment does not act as a confounder between covariates and target.": Can you elaborate how it ensures this?**
>
>  That is a good point, actually we require something weaker than 'not-confounding', so we change the sentence into "it ensures that the noise distribution is the same in all environments, which is a crucial property we test for within our methodology.
>
> **The definition of causal parents** $S^*$ **(near equation (1)) is circular:**
>
>  As we state under Equation (1), we consider this equation an SCM in the sense of Pearl (2016), who defines direct causal parents as "A variable X is a direct cause of a variable Y if X appears in the function that assigns Y’s value". Under this definition we believe that $S^*$ is well-defined. To emphasize this we will change $Y^e=X^e \beta^e+\varepsilon^e$ to $Y^e:= X^e \beta^e+\varepsilon^e$. However, with respect to Assumption 1 your point is certainly valid and we change "There exists a subset $S^* \subseteq [D],$..." to, "Let $S^* \subseteq [D]$ be defined as above,..."
>
> **below Lemma 1, "the above null hypothesis": which one:**
>
>  We will specify that we mean $H_{0,S}$.
>
> **Assumption 2, "where $f_d$ is a polynomial of finite  but otherwise arbitrary": I'm not quite sure what this means. Is it that if
>  $y$ is a parent, then $f_d(.)=...$**
>
>  Thank you for bringing this up. Indeed what you wrote is precisely it and we will make it explicit in the paper. In the proof we evaluate the variance of the residuals in two environments, and the polynomial relationship ensures that the variance is a ratio of two polynomials with respect to the entries in $\beta^e$. This gives rise to a polynomial equation which can at most be solved for a finite number of $\beta^e$ and thus for the proof we don't need an explicit form.

---

### Meta-Review · Area_Chair_ybn3 · 2024-04-21

The authors consider the problem of identifying the parents of a target variable Y from a set of variables X, in an observational study setting, where we may have access to multiple environments. The proposed method is mainly based on invariant causal prediction (ICP, Peters et al. 2016). Compared with ICP, the proposed method allows the causal mechanism (i.e., coefficients) of Y to be different across different environments. The authors provide theoretical guarantee of their proposed algorithm, and test the performance of their algorithm on synthetic data. The has been a robust discussion.